Resource

# *Leishmania*-infected macrophages release extracellular vesicles that can promote lesion development

Anna Gioseffi[1,*], Tim Hamerly[2,*], Kha Van[1], Naixin Zhang[1], Rhoel R Dinglasan[2], Phillip A Yates[3], Peter E Kima[1]

*Leishmania donovani* **infection of macrophages results in quantitative and qualitative changes in the protein profile of extracellular vesicles (EVs) released by the infected host cells. We confirmed mass spectrometry results orthogonally by performing Western blots for several *Leishmania*-infected macrophage-enriched EVs (LieEVs) molecules. Several host cell proteins in LieEVs have been implicated in promoting vascular changes in other systems. We also identified 59 parasite-derived proteins in LieEVs, including a putative *L. donovani* homolog of mammalian vasohibins (LdVash), which in mammals promotes angiogenesis. We developed a transgenic parasite that expressed an endogenously tagged LdVash/mNeonGreen (mNG) and confirmed that LdVash/mNG is indeed expressed in infected macrophages and in LieEVs. We further observed that LieEVs induce endothelial cells to release angiogenesis promoting mediators including IL-8, G-CSF/CSF-3, and VEGF-A. In addition, LieEVs induce epithelial cell migration and tube formation by endothelial cells in surrogate angiogenesis assays. Taken together, these studies show that *Leishmania* infection alters the composition of EVs from infected cells and suggest that LieEVs may play a role in the promotion of vascularization of *Leishmania* infections.**

## Introduction

In addition to secreted molecules, eukaryotic cells release membrane-enclosed vesicles (Kalra et al, 2012; Akers et al, 2013). Vesicles released by cells are subdivided into three categories that differ in their size, cellular origin, and molecular composition. Exosomes, the smallest of extracellular vesicles (EVs), range in size from 30 to 200 nm and originate from multivesicular compartments of the endocytic pathway (Akers et al, 2013), apoptotic bodies released by dying cells range in size from 50 to 5,000 nm, and microvesicles that are in the size range from 50 to 1,000 nm arise from budding and fission of the plasma membrane (Kalra et al, 2012). There are several reasons for the growing interest in the

characteristics and functions of exosomes including: (1) Evidence that exosomes from each cell type display a unique molecular composition that can be exploited to better characterize clonal tumors, for example, and monitor their metastatic progeny (Smith & Lam, 2018; Junqueira-Neto et al, 2019). (2) Exosomes have been implicated in cell-to-cell communications. Although the mechanistic details of how and where exosomes execute these functions is not fully understood, this characteristic is being exploited to deliver cell modulatory molecules to well described targets (Barile & Vassalli, 2017; Hardin et al, 2018). (3) Exosome content can be influenced by the environment and health of their cell of origin (de Jong et al, 2012; Panigrahi et al, 2018). For example, changes in oxygen availability could result in hypoxic conditions, which may influence the molecular composition of secreted exosomes (Kucharzewska et al, 2013). These functions can be exploited to identify exosome-derived biomarkers that can inform on the status of a disease or an infection using less invasive medical techniques (Zhang et al, 2016). (4) In infectious disease studies, there is evidence that exosomes from infected cells are composed of molecules that can act as immunomodulators or as potential vaccine candidates (Schorey et al, 2015; Shears et al, 2018).

The content and potential functions of exosomes derived from axenic *Leishmania* promastigotes have been reported (Silverman et al, 2008; Atayde et al, 2016). One outstanding question is whether infected cells that harbor *Leishmania* parasites, release parasite-derived molecules in their exosomal output. Hassani and Olivier (2013) showed that at least one parasite protein, leishmanolysin (gp63) is detected in exosomes recovered from macrophages infected with *Leishmania mexicana* parasites. However, it is important to appreciate that gp63 is a somewhat unique molecule. The Olivier laboratory had shown that upon infection of macrophages with promastigote forms, unlike most parasite molecules, gp63 is shed into infected cells where it is trapped within intracellular vesicles not associated with the *Leishmania* parasitophorous vacuole (Gomez et al, 2009; Gómez & Olivier, 2010). That finding was the impetus for the studies from the Olivier laboratory that led them to evaluate whether those gp63-containing vesicles could access the exosomal pathway in infected cells (Dong et al, 2019).

---

[1]Department of Microbiology and Cell Science, University of Florida, Gainesville, FL, USA [2]Emerging Pathogens Institute and Department of Infectious Diseases and Immunology, University of Florida, Gainesville, FL, USA [3]Department of Biochemistry and Molecular Biology, Oregon Health and Science University, Portland, OR, USA

Correspondence: pkima@ufl.edu
*Anna Gioseffi and Tim Hamerly contributed equally to this work

It is known that gp63 is significantly down-regulated and changes its location in the parasite as promastigotes transform to the amastigote form within infected macrophages (Yao et al, 2003; Hsiao et al, 2008). Considering this change in the localization of gp63 within the parasite, it is not known whether later stage macrophage infections, that harbor amastigotes forms, would continue to release gp63 in exosomes. Therefore, it remains unknown whether parasite molecules that are synthesized in amastigote (Hsiao et al, 2008) forms within macrophages in long-term infections are released in exosomes.

To address this question, we performed proteomic analyses of LieEVs that were released from established (>72 h) *Leishmania donovani* infections of RAW264.7 macrophages. We identified host- and parasite-derived molecules that may mediate *Leishmania* pathogenesis and evaluated the potential biological function of specific LieEV molecules during macrophage infection.

## Results

### Isolation and characterization of EVs released from *L. donovani*–infected RAW264.7 macrophages

In the studies discussed here, RAW264.7 macrophage infections were initiated by incubation of macrophages with metacyclic promastigotes for 24 h, after which the cultures were washed to remove extracellular parasites. Cultures were examined microscopically to ensure that uninternalized parasites were thoroughly

removed (Fig S1). Infected cultures were then replenished with complete medium prepared with exosome-depleted serum and cultured for an additional 48 h to evaluate long-term infected cells. Exosomes in the culture medium were isolated by differential centrifugation, which includes two low-speed centrifugations, filtration of the culture medium through a 0.22-*µ*m filter, and finally, two rounds of ultracentrifugation at 100,000*g* for 3 and 18 h, respectively (Fig 1A). Considering that this method cannot exclude apoptotic vesicles or microvesicles that are below ~200 nm, these preparations are labeled as *Leishmania* infection exosome–enriched EVs (LieEVs) or control cell exosome–enriched EVs (ceEVs).

The EV preparations were analyzed by NanoSight Tracking Analysis (NTA) to estimate overall quantity and size of vesicles. The average size of vesicles in ceEVs was 162 ± 28 nm, compared with 175 ± 21 nm for LieEV, which although slightly larger was not statistically significant (Fig 1B). However, the concentration of particles in the LieEV was significantly greater than that in ceEVs after 48 h (1.5 × 10⁵ versus 1.0 × 10⁵ mean particles/cell, *P* = 0.0082).

EV preparations were also examined by electron microscopy to evaluate gross morphology. The micrograph in Fig 1C shows representative images from scanning electron microscopy (SEM) of LieEVs. The EVs were observed to have rounded appearance that is characteristic of exosomes (Dreyer & Baur, 2016). EVs were also evaluated by transmission electron microscopy (TEM) where they were observed to have a cup shaped appearance (Fig 1D). Size estimates of EVs from TEM and SEM were ~79 ± 35 nm. Although the sizes are somewhat smaller than the size estimates by NTA (could be due to the protocol for sample processing), this size range is consistent with prior descriptions of exosomes (Singh et al, 2012;

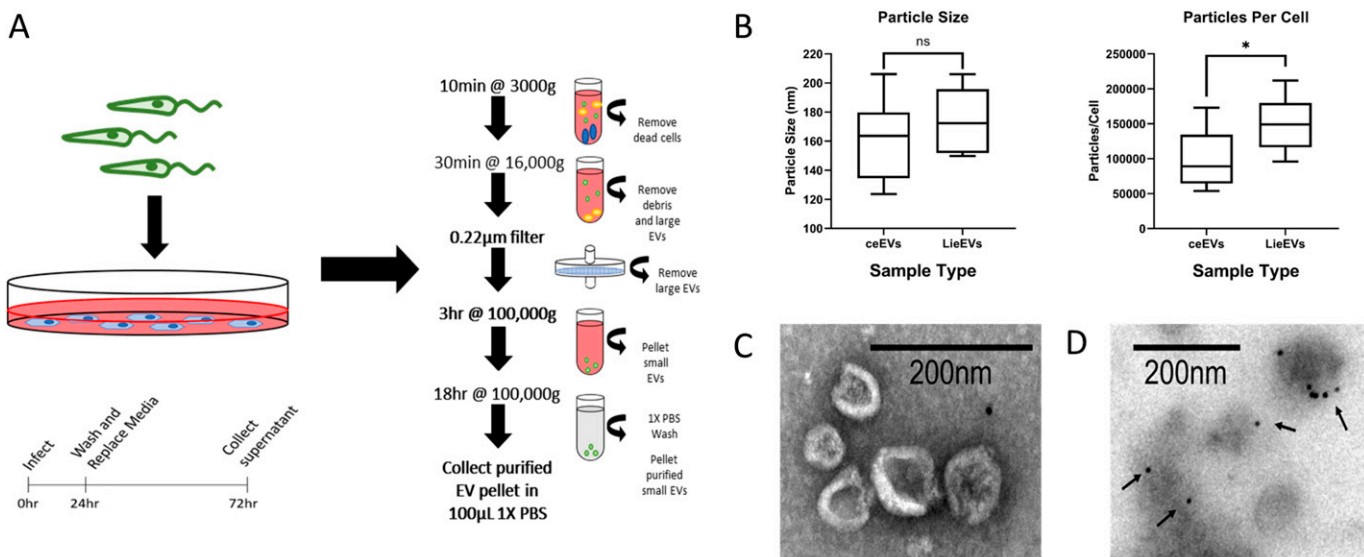

**Figure 1. Isolation and characterization of extracellular vesicles from *Leishmania donovani*–infected macrophages.**
**(A)** Workflow for collection of extracellular vesicles from RAW264.7 macrophages infected with *L. donovani* parasites. Infection was performed in media supplemented with exosome-depleted serum. After 24 h, the culture medium was removed. Cultures were washed to remove uninternalized parasites and replenished with fresh medium supplemented with exosome-depleted serum. After an additional 48 h, culture medium was recovered, pooled, and processed following the centrifugation and filtration steps shown in the figure. **(B)** Nanoparticle tracking analysis was performed from which particle size distribution and particle concentration was obtained. Plot of particles/cell was calculated using cell count at the end of the infection. Data for graphs were obtained from multiple experiments (ceEV n = 7, LieEV n = 7; *P = 0.0082). **(C)** Representative image of vesicles in LieEV preparation processed for scanning electron microscopy. **(D)** Representative transmission electron microscopy image of immunogold CD9-labeled particles in LieEVs. Arrows point to gold particles denoting reactivity of antibody.

Raposo & Stoorvogel, 2013; Yuan et al, 2017; Bachurski et al, 2019). To initially confirm that the EV preparations contained expected exosomal molecules, vesicles in the LieEV (and ceEVs) preparations were labeled with anti-CD9 and evaluated by immuno-EM. CD9 is a tetraspanin that has been widely shown to be a marker of exosomes (Kowal et al, 2016). Fig 1D shows representative vesicles decorated with gold particles denoting CD9 positivity.

### Differential host protein composition of LieEVs

We determined the proteomic composition of LieEVs recovered from infected RAW264.7 murine macrophages and compared them to ceEVs from uninfected cells. A common concern with proteomic profiling is the likelihood that low-abundance proteins can be masked either by highly abundant molecules in the preparation or by "contaminants" that can include lipids, which are a major component of the multivesicular bodies from where the vesicles originated. To minimize this potential limitation and increase reproducibility, we used in-gel digestion approaches followed by liquid chromatography tandem mass spectrometry (LC–MS/MS)

(Ji et al, 2013; Schey et al, 2015; Smith et al, 2016). We identified 618 murine proteins (Table S1) from six replicate LieEV and ceEV samples (minimum of two peptides and a false discovery rate [FDR] < 5%). Of the 618 proteins, 459 were shared by both LieEVs and ceEVs (Table S1 and Fig 2A). The shared protein list included known EV markers: tetraspanins CD9, CD81, CD63, Annexins A1, A2 and A5, the programmed cell death 6–interacting protein (ALIX), and HSP70. Annexin A3 was among the 76 proteins that were found to be enriched in LieEVs as compared with ceEVs.

To orthogonally confirm the LC–MS/MS results, both LieEV and ceEV preparations were analyzed by Western blot for the presence of the exosome protein markers CD9 and CD63 as well as Annexin A3 (Fig 2C). Approximately $1 \times 10^{10}$ particles of LieEVs and ceEVs from three separate EV isolations were analyzed. They were evaluated alongside 50 μg each of representative whole cell lysates of uninfected and 72 h infected RAW264.7 macrophages. A Ponceau S stain of a representative blot confirms that the total protein loading of EV samples was comparable (Fig 2B). Whole cell lysate samples that were loaded based on protein estimated by bicinchoninic acid also had comparable total protein levels. CD9 appeared to be more

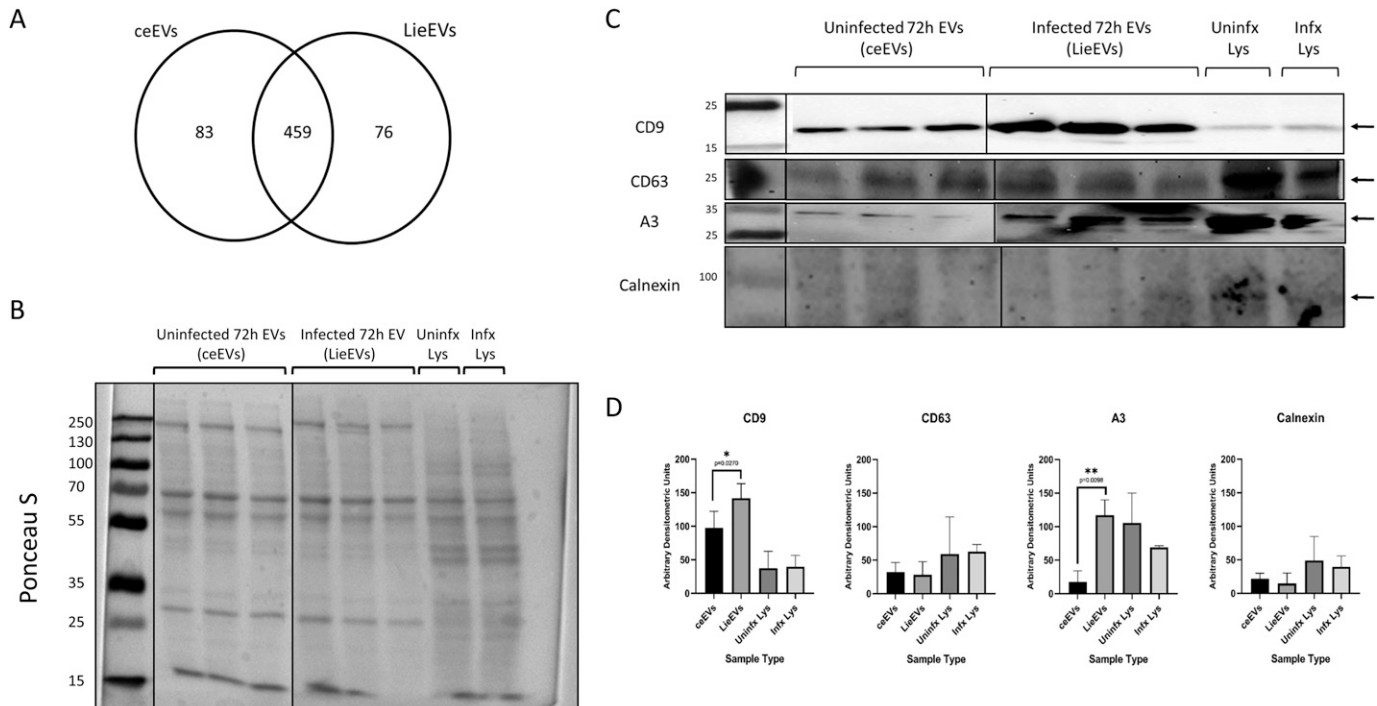

**Figure 2.   Mouse derived molecules in the LieEV proteome, includes known exosome markers as well as molecules implicated in mediating vascular changes in other systems.**
The proteome of LieEVs recovered after 72-h infection was compared with the proteome of ceEVs from uninfected cells. **(A)** Venn diagram was plotted using proteins identified by mass spectrometry and partitioned according to sample type. The presence and absence of host molecules with and without infection and common to both samples are indicated. **(B)** The protein content of the preparations was revealed by Ponceau S staining of the blots to normalize for material loaded into each well for each sample type. Approximately $1 \times 10^{10}$ particles from EV preparations from three replicate experiments and 50 μg of lysates from infected cells at the 72 h infection point were analyzed. **(C)** Western blot was performed to confirm the presence of known and novel exosome markers. Uninfected macrophages were treated in an identical manner as infected samples. The blots were then probed with anti-CD9, stripped, and probed with anti-Annexin A3. Identical blots were probed initially with anti-CD63, stripped, and probed with anti-calnexin. **(D)** Quantification was performed by measuring the mean gray background area using ImageJ software. Background pixel density was subtracted from the inverse of each measurement to obtain relative quantification values. Analysis of the blots showed that CD9 was significantly more abundant in LieEVs than ceEVs and cell lysates (*P = 0.0261, n = 3), whereas levels of CD63 were comparable for all samples. Annexin A3 was significantly more abundant in LieEVs than in ceEVs (*P = 0.0123, n = 3). Calnexin was barely detected in EVs as compared with cell lysates. Statistical test for differences was by ANOVA.
Source data are available for this figure.

enriched in LieEVs than ceEVs. CD9 levels in lysates loaded with comparable protein amounts appeared to be less abundant than in EV preparations (Fig 2D). CD63 levels were similar in the EV preparations, but those levels were less than in whole cell lysates (Fig 2D). Annexin A3 is a molecule of interest as it has been implicated in the promotion of angiogenesis (Park et al, 2005). Mass spectrometry analysis of LieEVs had shown that Annexin A3 was detected in LieEVs as compared with ceEVs, where it was likely below our detection limits. The Western blots probed for A3 showed limited if any A3 in ceEVs, which confirmed the mass spectrometry results. Blots were then stripped and probed with anti-calnexin antibodies because it is now widely accepted that calnexin is absent in exosomes (Théry et al, 2001; Abels & Breakefield, 2016). Calnexin was barely detected in the LieEV and ceEV preparations and notably, calnexin was also not identified in LieEV and ceEV preparations by mass spectrometry. Taken together, the results of the proteomic profiling of LieEVs demonstrated that *Leishmania* infection of macrophages induces qualitative changes in macrophage-derived molecules in EVs. The functions of these macrophage-derived molecules in LieEVs are of interest because of their possible roles in mediating pathogenesis.

### Host-derived molecules in LieEVs have been implicated in roles that promote vascularization including angiogenesis

To obtain insight into the possible functions of host proteins that are preferentially released in EVs from infected cells, we analyzed our proteomic data using the Ingenuity Pathway Analysis (IPA) program. Proteins identified by statistical analysis in Scaffold to have a fold change greater than two and a *P*-value less than 0.05 in LieEVs as compared with ceEVs were analyzed by the Core Analysis of IPA (Krämer et al, 2014). The significant findings of the analysis are shown in Fig 3A. The molecules enriched in LieEVs are predicted to have overlapping effects on several broad categories of biological processes, including Molecular and Cellular Functions, Diseases and Disorders, and Physiological System Developmental and Function. Within the Physiological System Developmental and Function category, more specific functions that are predicted to be affected include, Embryonic Development (14 identified molecules), Reproductive System Development (7 molecules) and Cardiovascular System Development and Function (14 identified molecules). The subset of molecules that were predicted to affect Cardiovascular System and Development were predicted to either increase or decrease vascular changes, including adhesion of endothelial cells, endothelial cell movement, and angiogenesis (shown in Fig 3B).

Together, these analyses suggested that molecules that are preferentially released in LieEVs could exercise several functions in the host, including changes in vascularization. Considering the molecular interactions and biological pathways associated with proteins in LieEVs as revealed by the IPA, we proceeded to investigate whether *Leishmania* infection induces the release of molecules in EVs that can promote changes in the vasculature that favor lesion development.

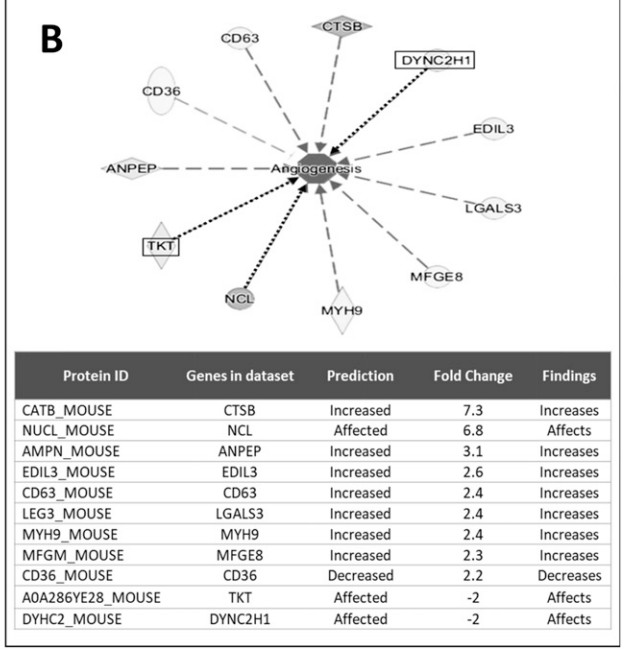

**Figure 3. Analysis of the host proteome by Ingenuity Pathway Analysis suggested that murine host proteins in LieEVs may differentially affect biological pathways and functions.**
**(A)** Subset of LieEV molecules classified in the top 5 "Molecular and Cellular Functions," top 5 "Diseases and Disorders" and top five Physiological systems development and function. **(B)** Most of the molecules in LieEVs that were categorized under cardiovascular system development have been implicated in the control of angiogenesis. Only those proteins that had a fold change >2 and *P*-value < 0.05 were analyzed. Canonical pathways were determined by using a right-tailed Fisher exact test by ingenuity pathway analysis.

## EVs from infected macrophages includes a putative homolog of mammalian vasohibin

As discussed earlier, the question of whether parasite molecules that are derived from long-term infected cells are released in EVs is still unresolved. To minimize the inclusion of exosomes derived from early-stage infection (less than 24 h post-invasion) in the analysis, infected cultures were washed after 24 h and replenished with medium supplemented with exosome-depleted serum. We were careful to make sure that free parasites were removed by washing. After an additional 48 h (72 h post-invasion), EVs from infected cultures were collected as described above. In addition to the *In-gel* processing of samples for mass spectrometry, samples were also processed by an *In-solution* protocol, to enable the identification of parasite molecules that are released in LieEVs, but that may be in low abundance. Studies on other infectious disease systems, including studies on exosomes derived from macrophages infected with *Mycobacteria* (Singh et al, 2012), showed that both methodologies can result in the identification of distinct sets of pathogen derived molecules in exosomes. A total of 59 parasite proteins were identified (two minimum peptides and FDR < 5%) in

LieEVs, using both methods. Tables 1 and 2 show 21 and 36 proteins were identified by *In-gel* and *In-solution* methods, respectively. Only a handful of proteins were identified from both digestion methods. This could be due to several factors including the difference in solubility and hydrophobicity of proteins and their relative abundance. Overall, 25 hypothetical proteins were identified. Both methods detected a ubiquitin ligase and RNA helicases. *Leishmania* RNA helicases have been described (Barhoumi et al, 2006). Interestingly, RNA polymerase II, which has been found in a complex with helicases, was also identified in LieEVs (Martínez-Calvillo et al, 2007). GP63, a protein that has been shown to be within exosomes from cells infected for up to 24 h, was not identified in these LieEV preparations. This was not unexpected as the expression of gp63 is not only down-regulated during the transformation of promastigote forms to amastigote forms but its localization switches from the parasite surface to an intracellular site (Hsiao et al, 2008), which could limit its potential to be exported from older *Leishmania* parasitophorous vacuoles.

Considering that mouse derived molecules in LieEVs are predicted to be associated with vascularization, LdBPK_270470.1.1, a parasite-derived protein from the LieEV proteome caught our

**Table 1.** Total parasite proteins identified in LieEVs using in-gel digestion approach for mass spectrometry.

| Parasite proteins in LieEVs (*In Gel* approach) | |
|---|---|
| **Identified proteins** | **Accession number** |
| 40S ribosomal protein S18, putative | LdBPK_360990.1.1 |
| ATP-dependent RNA helicase, putative | LdBPK_354080.1.1 |
| Calpain-like cysteine peptidase, putative | E9BET4_LEIDB |
| DNA repair and recombination protein RAD54, putative | LdBPK_240770.1.1 |
| DNA-directed RNA polymerase II subunit 2, putative | LdBPK_310170.1.1 |
| Dynein heavy chain (pseudogene), putative | LdBPK_272460.1.1 |
| Dynein heavy chain, putative | LdBPK_343990.1.1 |
| Hypothetical protein, conserved | LdBPK_060690.1.1 |
| Hypothetical protein, conserved | LdBPK_171000.1.1 |
| Hypothetical protein, conserved | LdBPK_343350.1.1 |
| Hypothetical protein, conserved | LdBPK_341580.1.1 |
| Hypothetical protein, conserved | LdBPK_364360.1.1 |
| Hypothetical protein, conserved | LdBPK_041100.1.1 |
| Hypothetical protein, unknown function | LdBPK_161290.1.1 |
| Hypothetical protein, unknown function | LdBPK_333050.1.1 |
| Inner arm dynein 5-1 | LdBPK_261000.1.1 |
| Kinesin, putative | LdBPK_190700.1.1 |
| Nucleoside transporter 1, putative | E9BTR3_LEIDB |
| Plasma membrane ATPase | E9BDY2_LEIDB |
| Proteasome regulatory non-ATPase subunit, putative | LdBPK_321260.1.1 |
| Protein kinase, putative | LdBPK_200970.1.1 |
| Protein kinase, putative | LdBPK_190590.1.1 |
| Ubiquitin ligase, putative | LdBPK_090790.1.1 |

TritrypDB ID and protein name are shown. Proteins with a minimum protein identification probability of 95% and two minimum peptides are shown. *P*-values were calculated using the Fisher exact test and spectral counts for four biological replicates (LieEV n = 4), where a minimum *P*-value was 0.05.

**Table 2. Total parasite proteins identified in LieEVs using in-solution digestion approach for mass spectrometry.**

| Parasite protein in LieEVs (*In solution* approach) | |
|---|---|
| **Identified proteins** | **Accession number** |
| 60S ribosomal protein L10, putative | LdBPK_040750.1.1 |
| AAA domain (Cdc48 subfamily), putative | LdBPK_301700.1.1 |
| Actin | LdBPK_041250.1.1 |
| α tubulin | LdBPK_130330.1.1 |
| ATP-dependent RNA helicase, putative | LdBPK_363150.1.1 |
| Dynein heavy chain, putative | LdBPK_231570.1.1 |
| Elongation factor 1-α | LdBPK_170170.1.α |
| Hypothetical protein, conserved | LdBPK_341260.1.1 |
| Hypothetical protein, conserved | LdBPK_321730.1.1 |
| Hypothetical protein, conserved | LdBPK_110310.1.1 |
| Hypothetical protein, conserved | LdBPK_252520.1.1 |
| Hypothetical protein, conserved | LdBPK_160990.1.1 |
| Hypothetical protein, conserved | LdBPK_030300.1.1 |
| Hypothetical protein, conserved | LdBPK_061130.1.1 |
| Hypothetical protein, conserved | LdBPK_301340.1.1 |
| Hypothetical protein, conserved | LdBPK_364490.1.1 |
| Hypothetical protein, conserved | LdBPK_060440.1.1 |
| Hypothetical protein, conserved | LdBPK_331100.1.1 |
| Hypothetical protein, conserved | LdBPK_260570.1.1 |
| Hypothetical protein, conserved | LdBPK_341920.1.1 |
| Hypothetical protein, conserved | LdBPK_041160.1.1 |
| Hypothetical protein, unknown function | LdBPK_080920.1.1 |
| Hypothetical protein, unknown function | LdBPK_131440.1.1 |
| Hypothetical protein, unknown function | LdBPK_271720.1.1 |
| Isoleucyl-tRNA synthetase, putative | LdBPK_365870.1.1 |
| Phosphatidylinositol 3-kinase–like protein | LdBPK_020100.1.1 |
| Phosphatidylinositol kinase, putative | LdBPK_301840.1.1 |
| Phosphopantothenate–cysteine ligase, putative | LdBPK_251980.1.1 |
| Present in the outer mitochondrial membrane proteome 22-2 | LdBPK_333090.1.1 |
| Proteasome α 7 subunit, putative | LdBPK_110240.1.1 |
| Protein of unknown function (DUF3250), putative | LdBPK_231490.1.1 |
| Raptor N-terminal CASPase–like domain containing protein, putative | LdBPK_201280.1.1 |
| RNA helicase, putative | LdBPK_090090.1.1 |
| Serine/threonine protein phosphatase catalytic subunit, putative | LdBPK_280730.1.1 |
| Seryl-tRNA synthetase | LdBPK_110100.1.1 |
| Small GTP-binding protein Rab18, putative | LdBPK_331940.1.1 |
| Ubiquitin-protein ligase, putative | LdBPK_366600.1.1 |
| Vasohibin, putative | LdBPK_270470.1.1 |

TritrypDB ID and protein name are shown. Proteins with a minimum protein identification probability of 95% and two minimum peptides are shown. *P*-values were calculated using the Fisher exact test and spectral counts for three biological replicates (LieEV n = 3), where a minimum *P*-value was 0.05.

attention. This protein was annotated as a putative vasohibin in the *Leishmania* genome. Vasohibins have been characterized in mammalian cells (Du et al, 2017), where they play a role in angiogenesis. In mammals, there are two isoforms, Vash 1 and Vash 2, which are expressed by different cell types and have been ascribed antagonistic functions related to angiogenesis. Vash 2 expressed by mononuclear cells promotes angiogenesis, whereas Vash 1 expressed by endothelial cells inhibits angiogenesis. Mammalian vasohibins are also classified as tubulin tyrosine carboxypeptidases (Wang et al, 2019). In the *Leishmania* genome, there is a single putative vasohibin gene whose gene product is predicted to be larger by more than 300 amino acids, compared with mammalian vasohibins. An alignment of the putative *L. donovani* vasohibin with mouse Vash 1 and Vash 2 is shown (Fig S2). This *L. donovani* vasohibin (LdVash) has vasohibin-like domains, but its length suggests that it may have other functions that are yet to be determined.

To validate that LdVash is expressed in *Leishmania*-infected cells and released in LieEVs, two strategies were implemented. In the first strategy, recombinant *L. donovani* parasites were developed in which the *LdVash* gene was tagged at its endogenous locus with the mNeonGreen (mNG) fluorophore using the tagging scheme shown in Fig S3 (Tran et al, 2015). Gene targeting constructs encoding the mNG-*Thosea asigna* virus 2A peptide-puromycin resistance (PAC) genes as a contiguous polypeptide facilitated the simultaneous tagging of *LdVash* via homologous recombination and selection for integration via puromycin resistance (Fig S3A). The 2A peptide sequence mediates a co-translational "cleavage" event (de Felipe et al, 2006) that separates the PAC protein from the LdVash/mNG fusion. The advantage of this approach is that expression of the tagged *LdVash* gene is under control of the cognate *LdVash* gene pre-mRNA processing signals and regulatory sequences, making it more likely to exhibit endogenous expression and regulation. Analysis of a recombinant parasite line that was generated, confirmed that it contained the expected amplification product of 1,255 bp (Fig S3B and C).

Upon infection of RAW264.7 macrophages with the *L. donovani*/LdVash/mNG (LdVash/mNG+) lines, fluorescence expression in live, infected macrophages, confirmed that recombinant parasites within macrophages expressed LdVash/mNG (Fig 4A). The figure shows representative live infected cells after 24, 72, and 96 h postinfection. mNG fluorescence was limited to the parasite at 24 h. However, at 48 and 96 h, in addition to intense fluorescence expression by the parasite, dispersed fluorescence could be observed in more vesicular structures in the host cell. There was also visible mNG labeling of uninfected cells that were in proximity of infected cells (most apparent in 48 and 72 h samples). Cells infected with wild-type parasites (WT 48 h) had little to no detectable fluorescence. Similar observations were seen upon infection of primary macrophages in peritoneal exudates (PECs) from thioglycolate injected Balb/c mice (Fig S4). We proceeded to isolate LieEVs at 24- and 96-h post-invasion from RAW264.7 cultures infected with two LdVash/mNG+ parasites lines, then analyzed them by NTA. Fig 4B shows that after infection with these recombinant parasites, there was an increase in the total number of EVs overtime. Moreover, a subset of LieEVs were fluorescent (Fig 4C), presumably because of the release of LdVash/mNG in LieEVs. Confirmatory results that showed that LdVash is secreted in EVs were also obtained from NTA analysis of LieEVs from PECs infected with a LdVash/mNG+ line (Fig S4).

For the second strategy to validate that LdVash is included as cargo in EVs from infected cells, we generated rabbit antiserum to LdVash. The antiserum was raised to an internal peptide of LdVash that is not present in the sequence of mammalian vasohibin (highlighted in Fig S2). Western blots of LieEVs prepared from cells infected with LdVash/mNG+ parasites were positive for LdVash and LdVash/mNG (Fig 4D). Equivalent amounts of protein were analyzed in these blots as shown by a representative blot that was stained with Ponceau S (Fig S5). Densitometric analysis confirmed a reproducible increase in LdVash, LdVash/mNG expression in infected cell lysates and in LieEVs (Fig 4E). Together, these experiments provided orthogonal validation of the mass spectrometry results that indicated that parasite-derived Vasohibin-like protein (LdVash) is secreted in LieEVs.

## EVs derived from *Leishmania*-infected cells activate angiogenesis

Vascular changes in tissues, including angiogenesis, are the result of multiple cell-associated activities that can be evaluated by surrogate assays in vitro. To test the hypothesis that LieEVs contain proteins that can induce angiogenesis, we evaluated the capacity of LieEVs to induce cell migration, activate endothelial cells, and initiate endothelial cell tube formation. Cell migration is an important process in the formation of blood vessels as endothelial cells are recruited to the site of new blood vessel growth. The migration of the breast cancer epithelial cell line MDA-MB-231 is widely used to study the capacity of agents to promote cell migration (Kerbel et al, 2013). In a scratch assay, gap closure by MDA-MB-231 treated with EVs was measured every 2 h over a 20 h period. Gap closure by LieEVs was 92% complete, which was significantly greater ($P = 0.0019$ at 18 h, $P = 0.0001$ at 20 h) as than cells incubated with ceEVs where closure was 65% after 20 h (Fig 5A and B). Conversely, incubation with LieEVs that were disrupted by sequential rounds of sonication did not have any appreciable impact on the migration of the epithelial cell line (Fig 5C). The percentage of gap closure area was measured by Tscratch software (Gebäck et al, 2009).

Another surrogate assay of angiogenesis is the tube formation assay by the HUVEC line in a gel matrix in response to specific angiogenesis promoting stimuli. For these studies as well, a range of EV dosages per cell were evaluated for their capacity to promote tube formation of HUVECs. The results presented in Fig 5D and E show that as few as 5,000 LieEV particles/cell, but not ceEVs, induce significantly more tube formation as than normal media control. Tubes were evaluated by their total tube length (LieEV versus ceEV, $P < 0.0458$) as well as the number of branching points (LieEV versus ceEV, $P < 0.0458$) as an indicator of vessel complexity. Disrupted EVs reproducibly induce considerably less tube formation as than intact EVs, even though differences were not statistically significant. The effect on tube formation of suramin, which is a known inhibitor of tube formation is shown.

We next performed studies on HUVECs to determine whether incubation with LieEVs resulted in the release of angiogenesis promoting factors. The supernatant from HUVEC cell cultures that were incubated with EVs for 24 h was evaluated with a multianalyte 18-plex human angiogenesis panel (Thermo Fisher Scientific). In Fig 6, we show that 10,000 LieEV particles/cell as compared with an equal number of ceEVs induce the preferential release of several mediators, including IL-8 ($P = 0.005$), G-CSF ($P = 0.0012$) and VEGF-A ($P < 0.0001$)

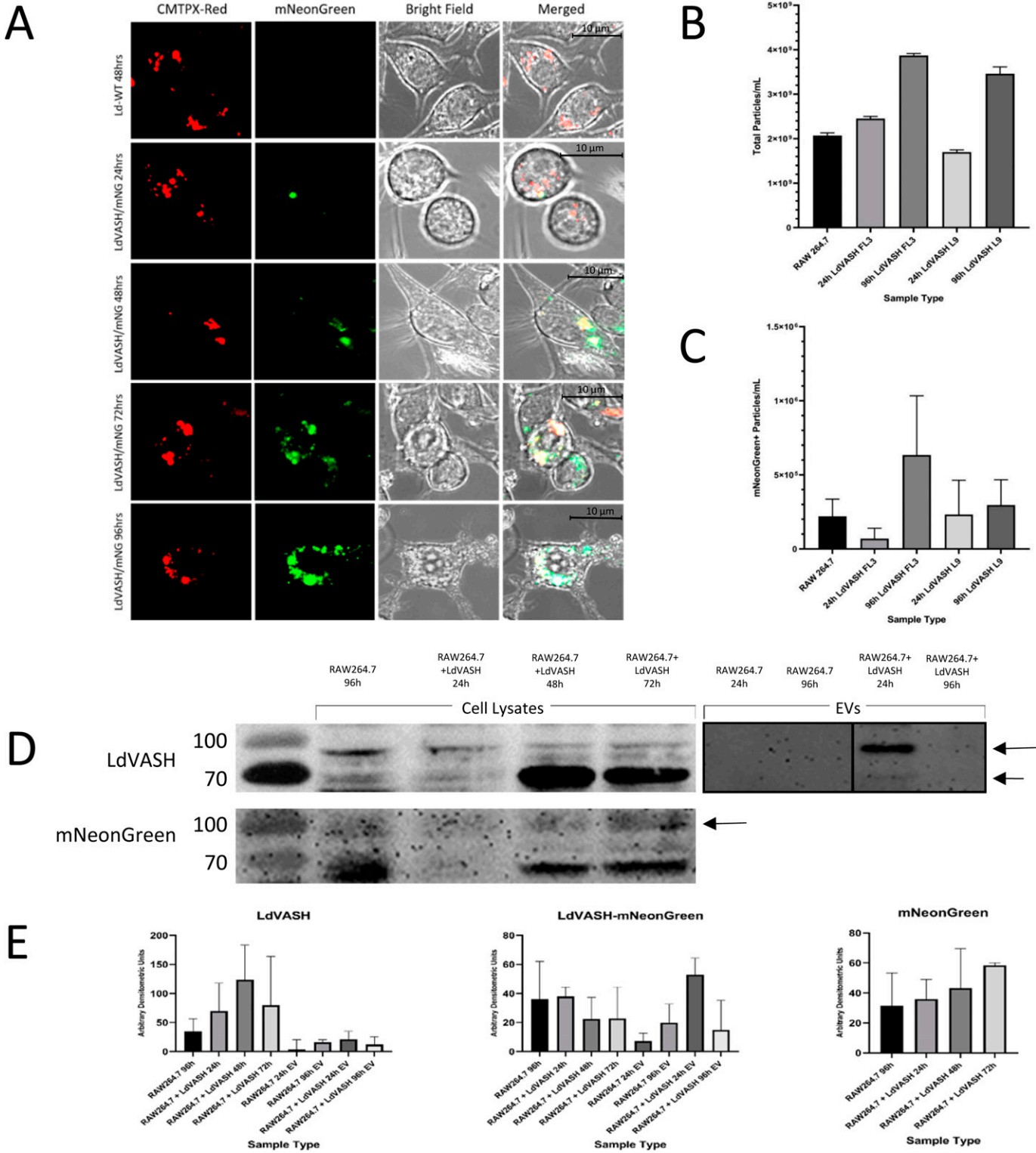

**Figure 4. The parasite homolog of vasohibin is expressed in *Leishmania*-infected cells and is a cargo in EVs from infected cells.**
Two parasites lines were derived in which the LdVash gene was endogenously tagged with mNeonGreen (LdVash/mNG+) (FL3 and L9). RAW264.7 macrophages were infected with the LdVash/mNG+ lines and analyzed for the distribution of mNG in infected cells and in EVs. **(A)** Representative images obtained from live infected cells at the indicated times. Metacyclic FL3 parasites were incubated with CMPTX-red before initiation of infection. White arrows point to parasites in the infected cells. Green label shows the distribution of LdVash/mNG as the infection progresses. These figures are representative of at least four live cell imaging experiments. **(B)** Particle analysis by NanoSight tracking analysis of total EVs recovered from uninfected cells or from cells infected with either of the two LdVash/mNG+ parasite lines at indicated times. **(B, C)** Enumeration of fluorescent EV particles by NanoSight tracking analysis in the samples in (B). Data in (B) and (C) were compiled from 1 representative of three experiments.

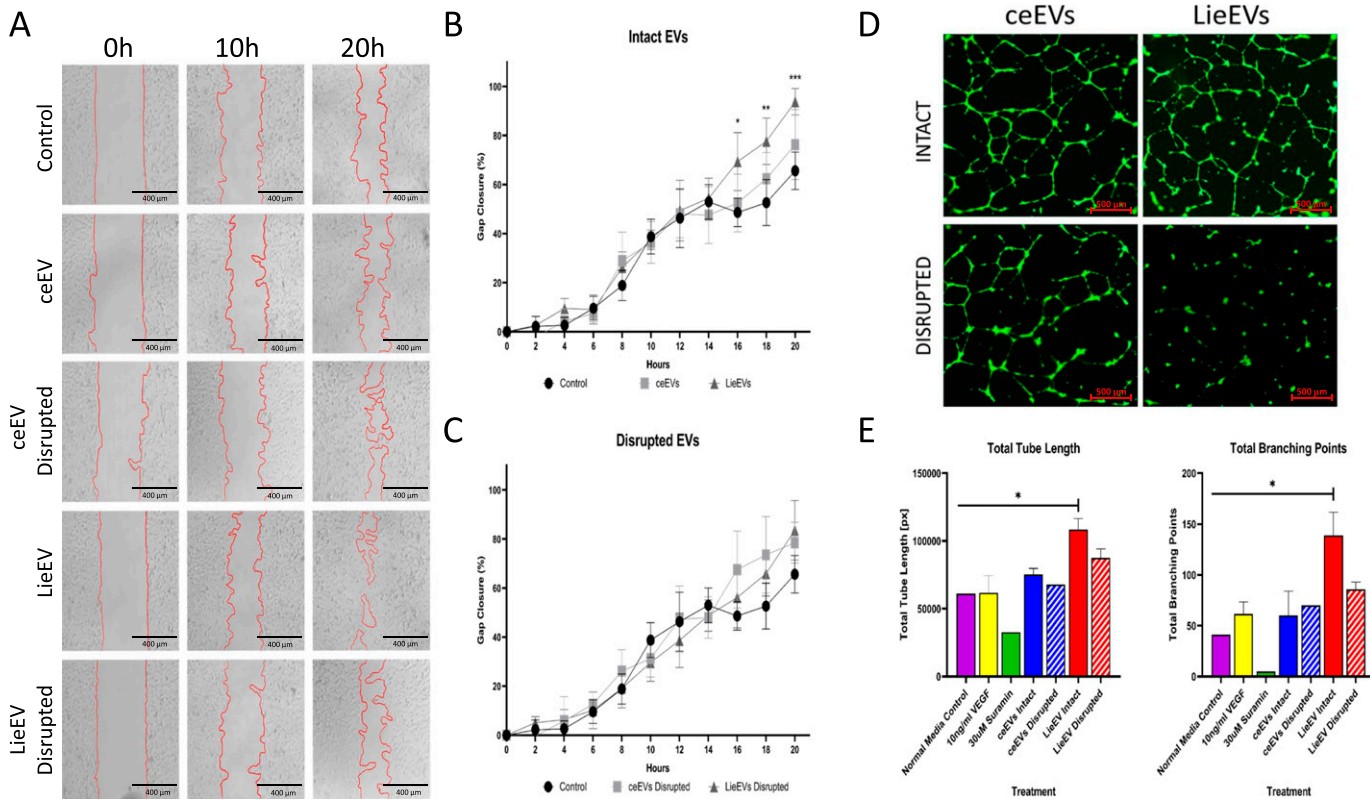

**Figure 5. LieEVs promote endothelial cell tube formation and cell migration.**
EVs from infected cells were evaluated alongside EVs from uninfected cells in surrogate assays of angiogenesis. **(A)** Representative images of gap closure at indicated times by the MDA-MB-231 epithelial cell in a scratch assay are shown. **(B)** Migration after incubation with intact LieEVs and ceEVs was captured over a period of 22 h and plotted. **(C)** Migration was also evaluated after incubation with disrupted LieEVs and ceEVs. Data were compiled from two biological repeats. **(D)** Capacity of LieEVs and ceEVs to promote tube formation by the HUVEC line was determined after incubation for 6 h. A representative image shows that intact LieEVs induce well organized tubes, which contrasts with either disrupted LieEVs or intact or disrupted ceEVs. **(E)** Changes in tube length and number of branching points induced by treatment with VEGF, LieEVs and ceEVs were plotted. Disrupted LieEVs and ceEVs and suramin that inhibits endothelial cell movement were also analyzed. For statistical analysis of differences, one-way ANOVA was performed with multiple comparisons for each time point from at least two biological repeats, followed by Brown–Forsythe's and Bartlett's tests.

from HUVEC cells. Secretion of other pro-angiogenic factors, including HGF, HB-EGF, Leptin, PDGF-BB, Follistatin, and Emmprin were also found to be significantly enhanced by incubation with LieEVs.

Taken together, we have provided evidence that *Leishmania*-infected macrophages release EVs that contain qualitatively different host-derived molecules as compared with uninfected macrophages. Some of the host cell molecules that are preferentially released in EVs from infected cells have been shown to promote changes in tissue vascularization in other biological contexts. Here, we provided evidence that LieEVs have the capacity to activate endothelial cells to secrete molecules that have been shown to promote angiogenesis and induce cell migration and the formation of tubes. In addition, we showed that LieEVs contain at least 59 parasite-derived molecules. Amongst the parasite-derived proteins, we identified LdVash, which is a putative homolog of mammalian Vasohibins that are known to contribute to angiogenesis.

Future in vitro and in vivo studies will shed more light on the functional characteristics of the LdVash and of other molecular components in LieEVs and their potential contributions to *Leishmania* lesion development.

## Discussion

*Leishmania* parasites live in phagocytic cells, wherein they replicate within membrane enclosed compartments. Like other eukaryotic cells, phagocytic cells including macrophages constitutively release EVs that play important biological functions during normal and disease states. The functions of exosomes and their cargo are of great interest, as their composition may indicate processes such as cancer metastasis or pathogen infection. Before these studies, it was not known whether proteins derived for *Leishmania* parasites

Increases in EVs released by older infections was significant (by ANOVA); however, presence of fluorescent EVs was reproducible but not statistically significant. **(D)** Western blot analysis of lysates and EVs from LdVash/mNG+ infected cells. Blots were probed with antiserum to LdVash and anti-mNG. Blots are representative of three experiments. **(E)** Densitometric scans of indicated bands from three experiments are shown. Statistical test for differences was by ANOVA.
Source data are available for this figure.

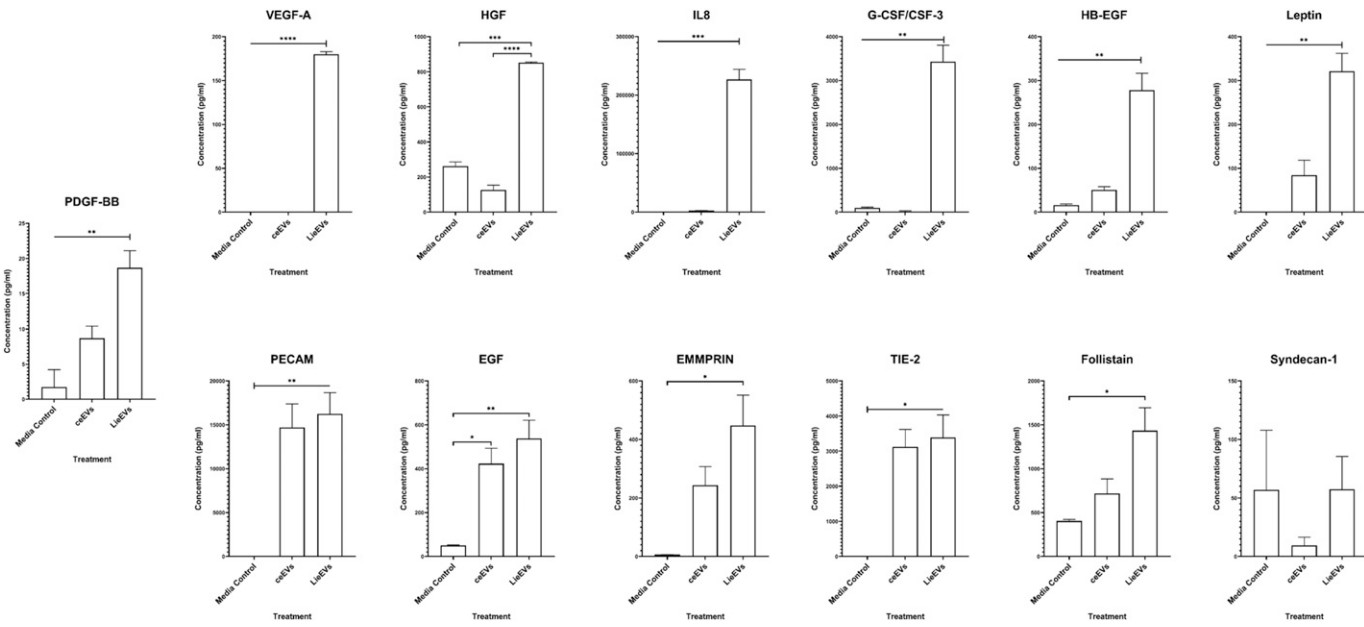

**Figure 6. LieEVs activate endothelial cells to release angiogenesis promoting cytokines.**
A multianalyte assay of secreted molecules implicated in angiogenesis was performed on supernatant fluid recovered after 24-h incubation of LieEVs or ceEVs with endothelial cells. Concentrations of each analyte were calculated from standard curves that were included in each experiment. Data were compiled from two biological repeats and analyzed using one-way ANOVA with multiple comparisons for statistical analysis.

could access the exosome secretion machinery of long-term infected cells. In this study, we have shown that at least 59 parasite-derived molecules are detected in the exosomal output of infected cells. We also reported that *Leishmania* infection induces the release of EVs that are qualitatively different from EVs that are released from uninfected cells. Analysis of the likely downstream functions of proteins that are preferentially released in the exosomal output by infected macrophages revealed that many of these molecules have been implicated in functions associated with changes in tissue vascularization in other systems. Based on these findings, we hypothesized that EVs from infected cells play a role in lesion vascularization.

The likelihood that molecules preferentially released in LieEVs, as compared with ceEVs, could play important roles in vascular changes including angiogenesis in *Leishmania* infections was intriguing. Horst et al (2009), had shown that cutaneous infection by *Leishmania major* parasites in the hind foot of C57BL/6 wild-type mice resulted in lesions that were extensively vascularized, as lymphatic and blood vessels were readily evident as the infection established. They then implicated the expression of carcinoembryonic antigen-related cell adhesion molecule 1 (CEACAM1) on mononuclear cells (CD11b[hi] cells) as essential mediators of angiogenesis in *L. major* infected lesions. Also performing studies with *L. major*, Weinkopff et al (2016), showed that there was increased expression of VEGF-A and VEGF receptor (VEGFR-2) at the site of infection that mirrored the increase in lesion size and parasite numbers. It is noteworthy that we found that LieEVs induce endothelial cells to secrete VEGF-A (Fig 6). In infections with *L. donovani*, where lesions form in visceral tissue, Yurdakul et al (2011), described vascularization and neovascularization of the red pulp and white pulp regions of the spleen, respectively. They attributed splenic vascularization to Ly6C+ inflammatory monocytes. In a more recent study, Dalton et al (2015) showed that neurotrophic tyrosine kinase receptor type 2 (Ntrk2,

also known as TrkB) was aberrantly expressed on splenic endothelial cells after *Leishmania* infections. They then showed that the ligand(s) for Ntrk2 were most likely expressed by macrophages in the infected spleen and that inhibition of signaling through Ntrk2 blocked white pulp neovascularization. It is noteworthy that none of these studies evaluated the specific contributions of infected cells to the vascular changes in lesions, which they described. Nonetheless, together, those studies suggested that there are changes in vascularization that may promote lesion development in leishmaniasis.

To provide support for the novel function of EVs in *Leishmania* infections, we evaluated LieEVs in surrogate assays of angiogenesis. These assays included the promotion of cell migration and induction of tube formation by an endothelial cell line, and the release of angiogenesis promoting molecules by endothelial cells after incubation with LieEVs. The development of new blood vessels has been shown to be necessary for wound healing, organ development and other processes that can facilitate cell trafficking in tissues (Carmeliet, 2003). LieEVs promoted greater epithelial cell migration and endothelial cell tube formation. In addition, LieEV activation of endothelial cells induced the preferential release of PDGFBB, VEGF-A, G-CSF/CSF-3, HGF, Leptin, and IL-8. A role for VEGF-A in *Leishmania* lesions was discussed above (Weinkopff et al, 2016). Also, it is noteworthy that IL-8 has been shown to be elevated in *Leishmania* lesions (Cáceres-Dittmar et al, 1993; Boussoffara et al, 2019). Our studies, therefore, suggest that cargo molecules in LieEVs have the potential to activate endothelial cells and to induce their release of potent mediators of vascularization at the lesion site that could promote angiogenesis. Future studies should attempt to distinguish the differential roles of parasite and host-derived molecules in the promotion of these endothelial cell responses.

Considering our hypothesis, we proceeded to investigate *Leishmania* Vasohibin, a putative homolog of mammalian Vasohibins, which was identified in LieEVs. Vasohibins (Vash) are molecules that play important roles in angiogenesis in mammals (Du et al, 2017), although the function of their *Leishmania* homolog was previously unstudied. The Vash gene has been identified in the genome of most eukaryotes, including the genome of mammals and the genomes of lower eukaryotic organisms (Sato, 2013; Du et al, 2017). In mice and humans, there are two variants of vasohibin, Vash 1 and Vash 2 that are encoded by two genes with distinct chromosomal locations. The Vash 1 gene is located on chromosome 14 and is preferentially expressed in endothelial cells, whereas the Vash 2 gene is located on chromosome 1 and is preferentially expressed by macrophages (Du et al, 2017). The genes share ~52% identity at the amino acid level and are each composed of 365 amino acids (VASH 1) and 355 amino acids (VASH 2) (Du et al, 2017). Vash 1 and Vash 2 have been shown to express contradictory functions in angiogenesis, Vash 1 is anti-angiogenic as compared with Vash 2, which is pro-angiogenic. Recent studies have suggested that Vash are part of a tubulin carboxypeptidase complex that may play other roles in cells including tubulin stabilization (Aillaud et al, 2017). In contrast to mammals, lower eukaryotes such as *Leishmania* parasites possess a single vasohibin gene (Du et al, 2017). In most sequenced genomes of *Leishmania* listed in Tritrypdb.org there is a single putative vasohibin gene located in chromosome 27. There is some variation in the size of the encoded polypeptide, which is predicted to be composed of 620–635 amino acids depending on the parasite species A recent report described the vasohibin gene in Trypanosomes and characterized their role in tubulin detyrosination during cell division (van der Laanet et al, 2019). However, it is not known whether both the pro-angiogenic and anti-angiogenic activities attributed to mammalian vasohibins are activities expressed by the *Leishmania*-derived molecule.

Endogenous tagging of Vash and the development of parasites that express LdVash/mNG confirmed that LdVash is released in EVs from infected host cells. The observation that LdVASH/mNG fluorescence could also been seen in some parasite-free macrophages in cultures infected with LdVASH/mNG *Leishmania* provides additional evidence that this molecule can access the exosome secretion machinery of the host cell. It is presently not known whether the 59 parasite molecules that were identified follow the same pathway for packaging and release in exosomes. It is noteworthy that some of these molecules have been identified in circulating immune complexes in *L. donovani*–infected individuals (Jamal et al, 2017). This suggests that they are released under physiological conditions and that they may be candidates for diagnosis of infections. Future studies will further characterize the roles that these molecules play in *Leishmania* pathogenesis including in the promotion of lesion development.

## Materials and Methods

### Parasite culture

*L. donovani* wild-type (MHOM/S.D./62/1S-CL2$_D$) was obtained from Dr. Nakhasi's lab (FDA) and cultivated in M199 media (M0393; Sigma-Aldrich) containing 15% FBS, 0.1 mM Adenosine, 0.1 mg/ml folic acid,

2 mM glutamine, 25 mM Hepes, 100 units/ml penicillin/100 $\mu$g/ml streptomycin (15140122; Gibco), 1× BME vitamins (B6891; Sigma-Aldrich), and 1 mg/ml sodium bicarbonate with pH 6.8 at 26°C.

### Mammalian cell culture

RAW264.7 macrophages were obtained from ATCC (TIB-71) and maintained in DMEM (10-013-CV; Corning) supplemented with 10% FBS and 100 units/ml penicillin/100 $\mu$g/ml streptomycin (15140122; Gibco) (complete DMEM) at 37°C in a humidified atmosphere containing 5% $CO_2$. MDA-MB-231 epithelial cells were obtained from Dr. Hendrick Luesch's lab (University of Florida) and cultivated in complete DMEM at 37°C in a humidified atmosphere containing 5% $CO_2$. HUVEC endothelial cells were acquired from ATCC (PCS-100-013) and maintained in vascular cell basal medium (ATCC PCS-100-030) supplemented with 0.2% bovine brain extract, 5 ng/ml rh EGF, 10 mM L-glutamine, 0.75 units/ml heparin sulfate, 1 $\mu$g/ml hydrocortisone hemisuccinate, 2% FBS, and 50 $\mu$g/ml ascorbic acid (Endothelial Cell Growth Kit-BBE PCS-100-040; ATCC) (complete endothelial cell media).

### Macrophage infections

RAW264.7 macrophages were plated on 100 mm culture dish containing sterile glass coverslips at a concentration of $5 \times 10^6$ cells per dish, and cells adhered overnight at 37°C with 5% $CO_2$ before infection with metacyclic promastigotes. To enrich for metacyclic parasites, peanut agglutination (PNA) protocol (Sacks & Melby, 2001) was performed on stationary-phase wild-type promastigote cultures. Briefly, 4-d-old cultures of *L. donovani* parasites were washed twice and resuspended in incomplete DMEM at a concentration of $2 \times 10^8$ parasites/ml. PNA was then added to the parasites at a final concentration of 50 $\mu$g/ml and incubated at room temperature for 15 min. The parasites were then centrifuged at 200$g$ for 5 min to pellet agglutinated parasites. The supernatant was then collected and the PNA-metacyclic parasites were washed twice, resuspended in complete DMEM, and counted for infection. Parasites were then added to macrophage dishes at a ratio of 20:1 (parasites:macrophage). After 24 h dishes were washed three times with 1× PBS with vigorous shaking to remove uninternalized parasites. Culture media was then replaced with complete DMEM supplemented with 10% exosome-depleted FBS. Infection was then allowed to continue for an additional 48 h at 37°C with 5% $CO_2$ in complete exosome-depleted macrophage media (72 h total infection time) (Exosome-depletion of Fetal bovine serum was prepared by centrifuging FBS at 100,000$g$ for 18 h at 4°C then the supernatant was removed and passed through a 0.22 $\mu$m Polyethersulfone [PES] syringe filter). For live cell imaging, metacyclic parasites were incubated in 1:1,000 dilution of CellTracker Red CMPTX dye (C34552; Invitrogen) for 30 min at 37°C as recommended by the manufacturer. The parasites were washed and then incubated with cells. Image capture was on a Zeiss Axio Observer 7 with Apotome2 for Axio Observer.

### Extracellular vesicle isolation

Media supernatant fluid was collected from cultures at 72 h post-infection and pooled from 10 plates. The supernatant fluid was then

centrifuged at 3,000*g* for 10 min, followed by 30 min at 16,000*g* to remove parasites and cellular debris. The supernatant was passed through a 0.22 *μ*m Polyethersulfone (PES) syringe filter. The filtrate was then centrifuged for 3 h at 100,000*g* to pellet EVs. The supernatant fluid was removed, and the pelleted vesicles were washed with 1× PBS and centrifuged for 18 h at 100,000*g*. Pelleted vesicles were resuspended in 100 *μ*l 1× PBS. EV preparations were used immediately or stored at −20°C for longer term storage.

## Nanoparticle tracking analysis

An NS300 NanoSight (ATA Scientific) fitted with an NS300 flow-cell top plate and a 405 nm laser was used. EV preparations were diluted in PBS 1:500 immediately before analysis and analyzed at a camera level of 10. The detection threshold for all samples was 10. The operating script used was TEMP 25, CAPTURE 60, DELAY 10, and REPEAT 2. A single analysis consists of three 60 s video captures. 1 mL of sample was loaded into the sample chamber before video recordings. The solution was manually advanced 100 *μ*l between static captures. Data were analyzed on the NTA software 3.0 (ATA Scientific) to determine the size and concentration of EVs. GraphPad Prism 8 was used to graph particle size and number per cell, expressed as the mean ± standard error. Unpaired *t* tests, followed by the F-test were then performed using GraphPad Prism 8 to determine statistical significance.

## Immunoelectron microscopy

Fresh EV preparations were rapidly frozen with an HPM 100 high-pressure freezer (Leica EM HPM100). The frozen EV samples were freeze-substituted in 0.1% uranyl acetate and 0.25% glutaraldehyde in acetone at −80°C for 2 d. After freeze substitution, the samples were warmed up to −50°C more than 30 h, washed four times with dry acetone at −50°C, and then embedded in HM20 acrylic resin (18174; Ted Pella) at −50°C. The resin was polymerized under ultraviolet light at −50°C for 36 h. All the freeze substitution, temperature transition, resin embedding, and UV polymerization were carried out in an AFS2 automatic freeze substitution system (Leica EM AFS2). The HM20-embedded samples were sliced into 100 nm thin sections that were placed on nickel grids, which were then immunogold-labeled with an anti-CD9 antibody (1/50 dilution vol/vol) (PB9930; Boster Bio) using a previously described method (Kang & Staehelin, 2008). The immunogold-labeled sections were post-stained with an aqueous uranyl acetate solution (2% wt/vol) and a lead citrate solution (26 g/Lead nitrate and 35 g/l sodium citrate), and data were acquired using a Hitachi TEM H-7000 operated at 80 kV. For SEM, EV samples were processed with the aid of a Pelco BioWave laboratory microwave (Ted Pella). Samples were washed in PBS, post-fixed with 1% buffered osmium tetroxide, water-washed, and dehydrated in a graded ethanol series 25%, 50%, 75%, 95%, 100%, and critical point dried (Bal-Tec CPD030; Leica Microsystems). Dried particles were mounted on carbon adhesive tabs on aluminum specimen mounts, and carbon coated (328 UHR Cressington; Ted Pella). Specimens examined with secondary electrons and backscatter electrons, and digital micrographs were acquired with a field-emission SEM (SU-5000; Hitachi High Technologies America).

## In-gel digestion

Isolated EVs were processed by in-gel digestion for LC–MS analysis. Approximately 1 × 10^{10} particles from each sample were resolved on a 4–20% gradient gel (4561096; Bio-Rad), then visualized by SimplyBlue SafeStain (LC6060; Thermo Fisher Scientific). Each lane was cut into eight slices of approximately the same size, then reduced, alkylated, and digested with 400 ng of trypsin overnight at 37°C. Digestion was quenched with 1% formic acid (FA) in 50 mM ammonium bicarbonate buffer/50% acetonitrile (ACN). Peptides were dried using a speed-vac and stored at −20°C. For LC–MS analysis, peptides were reconstituted in 3% ACN/0.1% FA.

## In-solution digestion

EVs (20 *μ*g) were solubilized using 0.1% RapiGest SF Surfactant (186001860; Waters) then precipitated using methanol/chloroform as described previously (Kummari et al, 2015). Samples were then resuspended with 6M urea buffer before being reduced and alkylated. The samples were then diluted with MilliQ water to reduce the urea concentration and digested with 400 ng of trypsin overnight at 37°C. The following day, the digestion was quenched using concentrated acetic acid. The peptides were then desalted using two C18 ZipTip Pipette tips per sample following the manufacturer's suggested protocol (ZTC18S096; Millipore), dried using a speed-vac, and stored at −70°C before analysis.

## LC–MS data acquisition

Peptides generated from both in-gel and in-solution digestion were injected onto an Acclaim PepMap 100 C18 trap column (75 *μ*m × 2 cm; Thermo Fisher Scientific) using an Agilent 1260 Infinity capillary pump and autosampler (Agilent Technologies). The autosampler was maintained at 4°C, the capillary pump flow rate was set to 1.5 *μ*l/min, and an isocratic solvent system consisting of 3% ACN/0.1% FA. After 10 min, the trap column valve was switched to be in-line with an Acclaim PepMap RSLC C18 analytical column (50 *μ*m × 25 cm; Thermo Fisher Scientific), using an Agilent 1290 Infinity II column compartment, kept at 42°C. Peptides were resolved on the analytical column using an Agilent 1290 Infinity II UHPLC with nanoflow passive split flow adapter, maintaining 200 *μ*l/min flow pre-split, and resulting in ~300 nl/min flow on the analytical column at the beginning of the run. A two-solvent system consisting of (A) water/0.1% FA and (B) ACN/0.1% FA was used, with a gradient as follows: 7% B at 0 min, ramping to 35% B at 15 min, then ramping to 70% B at 16, held at 70% B until 18 min, before returning to 7% at 19 min, and holding until the end of the run at 30 min, with a post run equilibration of 10 min. Eluted peptides were analyzed by an Agilent 6550 QToF mass spectrometer equipped with a G1992A nanoESI source (Agilent Technologies). The source parameters were as follows: drying gas temperature was set to 200°C, flow of 11 liters/min, a capillary voltage of 1,200 V, and fragmentor voltage of 360 V was used. Data were acquired in positive ion mode using data dependent acquisition, with an MS scan range of 290–1,700 m/z at 8 spectra/s, MS/MS scan range of 50–1,700 m/z at 3 spectra/s, and an isolation width set to narrow (~1.3 m/z). Maximum precursors per cycle was set to 10, with dynamic exclusion enabled after two spectra, and release time set to 0.5 min. Peptides were fragmented by collision induced dissociation using

$N_2$ gas and a variable collision energy depending on the precursor charge and m/z. Reference mass correction in real time was enabled, with lock masses of 299 and 1,221 m/z used.

## Data analysis

Data acquired for each analyzed sample was converted to Mascot Generic Format (.MGF) using the Agilent Data Reprocessor (Agilent Technologies). Database searching of .MGF files was done using three search engines; X! Tandem (Bjornson et al, 2008) and OMSSA (Geer et al, 2004) via SearchGUI (Barsnes & Vaudel, 2018) v3.3.16 (Compomics), and Mascot Daemon v2.2.2 (Matrix Science). Data were searched against a concatenated decoy FASTA file containing mouse and *L. donovani* strain BPK282A1 proteins downloaded from UniProt. Search results from all three engines were combined and analyzed using Scaffold 4 (v4.8.1; Proteome Software Inc.). Thresholds of 5% FDR protein, 1% FDR peptide, and two peptides minimum were set for protein identification. In Scaffold, fold change was calculated using total spectra with ceEVs as the reference category. The Fisher exact test was then used to calculate statistical significance, and a *P*-value of <0.05 indicated proteins with statistically significant changes in abundance. The Benjamini–Hochberg multiple testing correction was applied.

## Pathway, function, and network analysis of EV proteins

IPA software (IPA; version 24390178) was used for functional analysis of host-derived EV proteins with abundance altered upon wild-type *L. donovani* infection. Proteins were analyzed which had a fold change >2 and *P*-value < 0.05. Canonical pathways were analyzed by using a right-tailed Fisher exact test. The calculated significance represents the probability of association of proteins with the canonical pathway by random chance alone.

## Western blotting

Protein samples were resolved using a 10% SDS gel and transferred onto a nitrocellulose membrane. Membranes were blocked with 5% nonfat milk in 1× TBST. All primary antibodies were acquired from Boster Biological Biotechnology (Bosterbio) and used at a 1:1,000 dilution and secondary antibodies were acquired from Abcam and used at a 1:2,000 dilution. Antibody dilutions were prepared in 1× TBST with 1% nonfat milk. Antibodies to CD9, CD63, and Annexin A3 were obtained from Boster Biological (PB9930, M01080, and PB9420, respectively) and Calnexin was obtained from the Developmental Studies Hybridoma bank at University of Iowa (270-390-2-S). Primary antibody incubation was performed at 4°C overnight with rocking and HRP-conjugated secondary antibody incubation was performed at room temperature for 1 h with rocking. Proteins of interest were visualized using chemiluminescence on the Invitrogen iBright imaging system (Thermo Fisher Scientific). Quantification of blots was performed using ImageJ (1.46r). First, raw tagged image files (TIFs) of each blot were opened, and then the rectangle tool was used to select the bands of interest. Each lane was then selected separately in a frame, and the measurement tool was used to measure the mean gray value.

A similar area of the background was measured to use as reference for each blot. Pixel density data were then inverted, and the background value was subtracted to determine final relative quantification values. GraphPad Prism 8 was then used to graph the quantification data and perform a one-way ANOVA with multiple comparisons to determine statistical significance.

## Epithelial cell migration assay

MDA-MB-231 cells were suspended in complete DMEM at $5 \times 10^5$ cells/ml, and 70 $\mu$l of cell suspension was added into each well of an Ibidi two-well culture insert in 35 mm $\mu$-Dish (81176; Ibidi), and cells were incubated at 37°C and 5% $CO_2$ for 24 h to adhere. Media were then replaced with exosome-depleted complete DMEM supplemented with $10^{10}$ EVs/ml and allowed to incubate an additional 2 h. Three replicates each of EVs from uninfected (ceEVs) and *Leishmania*-infected (LieEVs) at 20,000 particles/cell were evaluated in each experiment. Disruption of EVs was accomplished by sonication (6× pulses 30 s each). Cell culture inserts were then removed using sterile tweezers, creating a cell free gap, and images were taken every 2 h for 20 h (time points: 0, 2, 4, 6, 8, 10, 12, 14, 16, 18, and 20 h post–insert removal) using transmitted light on an AMG EVOS fl LED fluorescent inverted microscope; eight images were taken of the gap for each dish. Images were then processed and analyzed using TScratch software (https://www.cse-lab.ethz.ch/software; version 1.0). Thresholds were manually adjusted for each image to ensure proper gap measurement. GraphPad Prism 8 was used to graph the data, presented as the mean ± standard error. To determine statistical significance, GraphPad Prism 8 was used to perform one-way ANOVA with multiple comparisons for each time point from at least two biological repeats, followed by Brown–Forsythe's and Bartlett's tests.

## Tube formation assay

Tube formation assays were performed using BioVision's angiogenesis assay kit (K905-50; BioVision) and the manufacturer's suggested methods. Briefly, 50 $\mu$l of thawed extracellular matrix solution was added to each well of a sterile, pre-chilled 96-well cell culture plate on ice. The plate was rocked gently to evenly distribute the solution, then incubated at 37°C for 1 h to allow the solution to gel. Confluent flasks of HUVEC cells were then harvested using 0.25% trypsin and 2.21 mM EDTA (25-053-CI; Corning), washed, and resuspended at $4 \times 10^5$ cells/ml in exosome-depleted complete endothelial cell media. Fifty microliters of exosome-depleted complete endothelial cell media, supplemented with $1 \times 10^9$ intact or disrupted ceEVs or LieEVs per mL, 20 ng/ml VEGF or 60 $\mu$M suramin and 50 $\mu$l of resuspended cells ($2 \times 10^4$ per well) were then added to each well, and incubated for 6 h at 37°C with 5% $CO_2$ (5,000 particles/cell was lowest particle number that provided unambiguous results). After incubation, the medium was removed using a pipette without disturbing the cells or extracellular matrix. Wells were then carefully washed using 100 $\mu$l of provided wash buffer to remove any remaining FBS. Staining dye working solution was prepared by diluting staining dye concentrate 1:200 in wash buffer. Then, 100 $\mu$l of diluted staining solution was added to each well and incubated at 37°C for 30 min. Tube formation was then imaged

using light and fluorescent microscopy (FITC filter), and images were acquired. Images were then analyzed using WimTube (Wimasis Image analysis). The resulting quantitative data were graphed using GraphPad Prism 8, and one-way ANOVA with multiple comparisons was performed, followed by Brown–Forsythe and Bartlett's tests, to evaluate statistical significance.

### Endothelial cell activation and cytokine quantification

HUVEC cells were plated at a concentration of $1 \times 10^6$ cells per well of a six-well cell culture plate and allowed to adhere for 24 h in complete endothelial vascular cell basal medium at 37°C and 5% $CO_2$. Cells were then incubated in 1 ml endothelial cell media prepared with exosome-depleted serum which contained ceEVs or LieEVs at $1 \times 10^{10}$ EV/ml or 10,000 particles/cell. After 24 h at 37°C and 5% $CO_2$ the media supernatant was collected, centrifuged at 13,000$g$ for 10 min to pellet cellular debris, then aliquoted and frozen at –70°C prior to analysis. Detection of chemokines and cytokines was carried out with a premixed multianalyte magnetic Luminex assay (Angiogenesis 18-Plex Human ProcartaPlex Panel; EPX180-15806-901; Thermo Fisher Scientific) according to the detailed protocols by the manufacturer.

### Generation of recombinant Ld Vash/mNeonGreen parasites

The 5′- and 3′-targeting sequences for tagging LdVash with mNEON green at its endogenous locus via homologous recombination were PCR amplified from *L. donovani* genomic DNA using the following primers:

Coding sequence (forward: gaGGCC ACCTA GGCC atgtcgga-taaagtgctggccattg; reverse: gaGGCC ACGCA GGCC ccggcgtggtgag-cacatg); 3′-UTR (forward: gaGGCC TCTGT GGCC agcgtcgcgttccgttgttc; reverse: gaGGCC TGACT GGCC gaagttgatgcattacggagacactcg). For the generation of the mNEON green-TAV2A-PAC cassette, the plasmid backbone, mNEON green cassette, and 5′- and 3′-targeting sequences were flanked by SfiI restriction sites (capital letters) with unique overhangs (underlined) and assembled via multi-fragment ligation as described in Fulwiler et al (2011). For transfection, the targeting cassette was excised from the plasmid backbone via SwaI digestion followed by gel purification, and electroporated into *L. donovani* promastigotes using the high-voltage protocol of Robinson and Beverley (2003). Immediately after transfection, the parasites were aliquoted into 96-well tissue culture plates and subjected to twofold serial dilution to allow isolation of independent clones. Puromycin was added the next day to select for integration into the LdVash locus. Integration was confirmed via PCR as depicted in Fig S2.

Infected cells were visualized live, at room temperature on a Zeiss Axio Observer Z1/7; 63× water immersion objective. Images were captured with an Axiocam 503 controlled by Zen Acquisition software. All the images are original in .czi files which were converted into .tiff files with 8 bit for image J processing. The Z-stack images are generated by stacking all the single planes without any changes from the original tiff file pool. The final image was made by z-project with maximum intensity.

### Statistical analysis of data

After data acquisition, data for in vitro angiogenesis assays were analyzed and graphed using GraphPad Prism 8. Data were first imported, then a one-way ANOVA with multiple comparisons was performed, followed by Brown-Forsythe's and Bartlett's tests to determine statistical significance. Similarly, densitometry data for Western blot quantification of protein levels were acquired from ImageJ as described above, and then GraphPad Prism 8 was used to generate bar graphs and perform a one-way ANOVA with multiple comparisons to determine statistical significance. Significance was determined by a $P$-value ≤ 0.05.

## Data Availability

The mass spectrometry proteomics data have been deposited to the ProteomeXchange Consortium via the PRIDE (Perez-Riverol et al, 2019) partner repository with the dataset identifier PXD021769.

## Supplementary Information

## Acknowledgements

We thank the staff at the Uiniversity of Florida Electron Microscopy facility for their assistance with EM experiments. We thank Dr. Mariola Edelmann for assistance with IPA. Research reported in this publication was supported by the National Institute of Allergy and Infectious Diseases of the National Institutes of Health under Award Number R56AI143293 to PE Kima. The content is solely the responsibility of the authors and does not necessarily represent the official views of the National Institutes of Health. Also, support was provided in part by the UF Emerging Pathogens Institute (RR Dinglasan) and UF College of Veterinary Medicine (RR Dinglasan) and R03 (AI137636) from NIAID to PA Yates.

### Author Contributions

A Gioseffi: data curation, formal analysis, validation, investigation, methodology, and writing—review and editing.
T Hamerly: data curation and writing—review and editing.
K Van: data curation, validation, investigation, and methodology.
N Zhang: data curation and methodology.
RR Dinglasan: data curation, methodology, and writing—review and editing.
PA Yates: data curation, methodology, and writing—review and editing.
PE Kima: conceptualization, formal analysis, supervision, funding acquisition, investigation, methodology, project administration, and writing—original draft, review, and editing.

### Conflict of Interest Statement

The authors declare that they have no conflict of interest.

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
