## [Reviewer comments · Life Science Alliance]

Life Science Alliance

Leishmania-infected macrophages release extracellular vesicles that can promote lesion development

Anna Gioseffi, Timothy Hamerly, Kha Van, Naixin Zhang, Rhoel Dinglasan, Phillip Yates, and Peter Kima

DOI: <https://doi.org/10.26508/lsa.202000742>

Corresponding author(s): Peter Kima, University of Florida

Review Timeline:

Submission Date:	2020-04-15
Editorial Decision:	2020-05-14
Revision Received:	2020-09-13
Editorial Decision:	2020-10-01
Revision Received:	2020-10-06
Accepted:	2020-10-08

Scientific Editor: Shachi Bhatt

Transaction Report:

May 14, 2020

Re: Life Science Alliance manuscript #LSA-2020-00742-T

Dr. Peter E Kima
University of Florida
Department of Microbiology and Cell Science University of Florida Building 981, Box 110700
Gainesville, FL 32611

Dear Dr. Kima,

Thank you for submitting your manuscript entitled "Leishmania-infected macrophages release extracellular vesicles that can promote angiogenesis of Leishmania lesions" to Life Science Alliance. The manuscript was assessed by expert reviewers, whose comments are appended to this letter.

As you will see, the reviewers point out that your conclusions are not supported by the data provided. They provide constructive input on how to address the issues they note, and we would thus like to invite you to submit a revised version of your manuscript to us. Importantly, all three reviewers point to missing controls that need to get included. For example, a control showing that EVs do not derive directly from the parasites needs to get included as well as loading controls and controls on EV purity. Further, reviewer #2 points out that your findings do not support a role for macrophage-derived EVs in chronic Leishmania infection, and the manuscript text therefore needs to get re-written and conclusions toned-down. Finally, this reviewer points out that the figures are of insufficient quality. I don't know whether there were some conversion issues or whether you downsized the figures prior to upload, but we agree with this concern and it needs to get addressed, too.

In our view these revisions should typically be achievable in around 3 months. However, we are aware that many laboratories cannot function fully during the current COVID-19/SARS-CoV-2 pandemic and therefore encourage you to take the time necessary to revise the manuscript to the extent requested above. We will extend our 'scooping protection policy' to the full revision period required. If you do see another paper with related content published elsewhere, nonetheless contact me immediately so that we can discuss the best way to proceed.

Please note that papers are generally considered through only one revision cycle, so strong support from the referees on the revised version is needed for acceptance.

Thank you for this interesting contribution to Life Science Alliance. We are looking forward to receiving your revised manuscript.

Sincerely,

B. MANUSCRIPT ORGANIZATION AND FORMATTING:

*****IMPORTANT:** It is Life Science Alliance policy that if requested, original data images must be made available. Failure to provide original images upon request will result in unavoidable delays in

publication. Please ensure that you have access to all original microscopy and blot data images before submitting your revision.***

Reviewer #1 (Comments to the Authors (Required)):

In the manuscript by Gioseffi et al. they isolated exosome-enriched extracellular vesicles (EV) from un-infected and *Leishmania donovani*-infected macrophages. Unlike previous studies evaluating EV released from *Leishmania*-infected macrophages, the authors extended the infection time and isolated EVs 48 hours post-infection of Raw264.7 cells. They evaluated the EVs for host and parasite proteins and performed some functional studies. In their analysis they identified a number of host proteins which were unique to EVs isolated from infected macrophages including proteins involved in the angiogenesis. The potential role for EVs, isolated from infected macrophages, to induce angiogenesis was confirmed using the scratch assay and the tube formation assay.

This is a well-designed study that provides additional information regarding the protein composition of EVs released from *Leishmania*-infected macrophages. It also defines a potential link between the increased angiogenesis observed at the site of a mouse *L. major* and *L. donovani* infection and the ability of ieEVs to induce secretion of angiogenic molecules by endothelial cells. They used complementary assays to demonstrate that ieEVs can induce an angiogenesis-like process in vitro. However, they did not show whether this was mimicked in vivo. Nevertheless, the studies are supportive of ieEVs as indices of vascularization. I also appreciate that they used two different isolation techniques to obtain material for MS, as the use of both methods increases protein coverage.

The one control that is missing is that they do not show that the macrophage culture media used for EV isolation does not contain extracellular parasites. If the culture media from infected cells have live *Leishmania* these could also be releasing EVs that would co-purify with host-derived EVs. Although this is unlikely, it is important to show an absence of *Leishmania* in the culture media used for EV isolation.

Reviewer #2 (Comments to the Authors (Required)):

Comments to the author:

In this study, Gioseffi et al. investigate the composition of extracellular vesicles (EVs) released from macrophages infected with *Leishmania*. The proteomic analyses demonstrated that infected macrophages release EVs with different compositions of proteins implicated in promoting vascular changes, such as Vash. They suggest that EVs from infected macrophages induce the release of angiogenesis molecules by endothelial cells and promotes epithelial cell migration and tube formation by endothelial cells in vitro. From these results, the authors propose that EV from infected macrophages promote vascularization in chronic *Leishmania* infections. Although the topic of this study is interesting and relevant to the disease, the conclusions are beyond what is shown by the data. The authors have used the appropriate methodology to isolate and physically characterize the EV (NTA and electron microscopy). However, the analyses to evaluate the role of EV in angiogenesis were only done in vitro, and whether they will have any role in chronic *Leishmania*

infection is unknown.

Specific comments:

- 1) Structural concerns of the manuscript: The results section needs to be modified, and the combination of the results and discussion is not helpful in this manuscript. Even more important, the figures are of extremely low quality, some of them are unreadable, and the figure legends do not contain sufficient information. For example, the abbreviations used are not described, the statistical analysis and the number of replicates were not indicated.
- 2) The purity of EV should be assessed.
- 3) EVs have different membrane compositions, depending on the cell from which they have originated. The authors cannot rule out the possibility of EVs being release directly from leishmania, instead of macrophage origin.
- 4) Figure 3: The western blot for the GP63 band is not clear in the Ldcen^{-/-}. A loading control should be included.
- 5) Figure 4D: When closely examining the western blot, the LdVash 24h EV band is stronger than LdVash 72h and 96h. This result contrasts with the images of Figure 4A, which show an increase in the LdVash expression with the time. Consider quantify the western blot bands.
- 6) Figure 5: The authors should use skin epithelial cells or spleen endothelial cells to test the effect of EV since these are the tissues related to Leishmania infection.
- 7) Figure 5: The cell lineage used in angiogenesis assays should preferably be murine since the assay involves EVs from the mouse.
- 8) Figure 5: CeEV Disrupted image in (A) is not a good representative figure of the graph (C).
- 9) Figure 5: The status of angiogenesis molecules in the supernatant fluid from HUVEC cells incubated with disrupted EVs should be provided as a control.
- 10) Figure 5E: It appears the positive control (VEGF) did not work.
- 11) The authors do not formally investigate whether LiEV promotes vascularization in chronic Leishmania lesions since the analyses were done in vitro. Thus, their conclusions are not justified.

Minor comments:

- 1) On what basis were 20 parasites per macrophage chosen?
- 2) Annexins expression levels are adjusted to the functional state of the cell. The discussion might benefit from some reflection on how Annexin A3 can be involved in macrophages infection and the EV release.
- 3) The Annexin A3 and GP63 data appears to have no connection with the main point of the paper.
- 4) "Protein concentrations determined by BCA agree with the NTA analyses that there is greater total protein content in the ieEV samples; however, ceEVs and ieEVs appear to contain equal

protein content per particle." It is data not shown? The authors should discuss why ceEVs and ieEVs appear to contain equal protein content per particle.

5) Overstate sentence: "Together, these studies demonstrated that there are impressive changes in vascularization of both visceral and cutaneous lesions that appear to be promoted by molecules that are released from infected macrophages in infected tissues." The mentioned papers (Horst et al, 2009, Weinkopff et al, 2016, Yurdakul et al, 2011 and Dalton et al, 2015) demonstrated that the molecules were released by macrophages present in infected mice. None of these papers has shown that infected macrophages release the molecules.

6) Figure 4: The results section is different from the figure order. The description of Figure 4B in the text is referring to 4C in the figure. The authors state that after infection with the recombinant parasites there was an increase in the total number of EVs over time but there is no statistical test information in the figure and the legend.

Reviewer #3 (Comments to the Authors (Required)):

Leishmania donovani is the causative agent of a potentially fatal visceral infection. In this manuscript, Gioseffi, et al. explored the composition and role of extracellular vesicles (EVs) released from infected macrophages. *Leishmania* infected macrophage EVs (LiEVs) contain several host proteins that can cause endothelial cells to migrate, engage in tube formation and release angiogenesis promoting factors such as IL-8, GCSF/CSF-3 and VEGF-A. Additionally, more than 50 *L. donovani* - derived proteins were identified in LiEVs, including a homolog of mammalian Vasohibins (LdVash), an angiogenesis promoting mediator. Taken all together, these results show that *L. donovani* infection can alter the composition of LiEVs and plays a role in vascularization during chronic disease. This is one of the first reports on leishmanial proteins found in host exosomes, and the first on *L. donovani*. The manuscript has good logical experimental flow and discussion of the results and implications of the paper. Minor issues are highlighted below:

Minor:

1. The present studies were performed using RAW264.7 macrophages. The authors should include a brief explanation about why they chose immortalized cells over primary cells such as bone marrow derived macrophages, which are more physiologically relevant
2. The authors mention Hassani and Oliver 2013 (PMID: 23658846). It would be interesting to discuss and draw a parallel between the proteins found in *L. donovani* and those previously found by Hassani and Oliver in *L. mexicana*-derived exosomes
3. Figures and Tables:
 - a. Table 1 cannot be found in the manuscript
 - b. Include a loading control in the western blots of figure 2B, 3B and 4D
 - c. Part of the content of Figure 2 and 3 might be better represented into tables, consider separating them up
 - d. In the legend of Figure 5 there should be a closed parenthesis after "E" instead of a period
4. Minor grammatical errors:
 - a. In the result section Differential host protein composition in LiEVs it says: "low abundant proteins". The word "abundant" should be substituted with "abundance"

b. In the result section Extracellular vesicles derived from Leishmania-infected cells activate angiogenesis there should be a period between "cell tube formation" and "Cell migration"

c. All the "et al." have to be italicized throughout the paper

5. Please consider moving part of Isolation and characterization of extracellular vesicles released from Leishmania donovani infected RAW264.7 macrophages to the methods section

Reviewer #1 (Comments to the Authors (Required)):

In the manuscript by Gioseffi et al. they isolated exosome-enriched extracellular vesicles (EV) from un-infected and Leishmania donovani-infected macrophages. Unlike previous studies evaluating EV released from Leishmania-infected macrophages, the authors extended the infection time and isolated EVs 48 hours post-infection of Raw264.7 cells. They evaluated the EVs for host and parasite proteins and performed some functional studies. In their analysis they identified a number of host proteins which were unique to EVs isolated from infected macrophages including proteins involved in the angiogenesis. The potential role for EVs, isolated from infected macrophages, to induce angiogenesis was confirmed using the scratch assay and the tube formation assay.

This is a well-designed study that provides additional information regarding the protein composition of EVs released from Leishmania-infected macrophages. It also defines a potential link between the increased angiogenesis observed at the site of a mouse L. major and L. donovani infection and the ability of ieEVs to induce secretion of angiogenic molecules by endothelial cells. They used complementary assays to demonstrate that ieEVs can induce an angiogenesis-like process in vitro. However, they did not show whether this was mimicked in vivo. Nevertheless, the studies are supportive of ieEVs as indices of vascularization. I also appreciate that they used two different isolation techniques to obtain material for MS, as the use of both methods increases protein coverage.

The one control that is missing is that they do not show that the macrophage culture media used for EV isolation does not contain extracellular parasites. If the culture media from infected cells have live Leishmania these could also be releasing EVs that would co-purify with host-derived EVs. Although this is unlikely, it is important to show an absence of Leishmania in the culture media used for EV isolation.

We thank reviewer 1 for the complimentary statements about our studies. The reviewer remarked on our study design that we think distinguishes this study. When *Leishmania* infections are initiated with promastigotes, internalized parasites transform into amastigote forms after 16 – 18hrs. This transformation is accompanied by some changes in expressed proteins and their levels of abundance. gp63, for example, is highly expressed in promastigotes. Upon transformation to the intracellular form, it becomes less abundant and even changes its localization in the parasite from the cell surface to the cell cytosol. Our objective was to evaluate the composition of proteins in EVs from older infections., which should be more representative of infected cells in chronic infections. We reasoned that by thoroughly washing off the media after 24 hours of infection and replacement with media that is supplemented with exosome depleted serum, we would minimize the contribution of uninternalized parasites to the overall EVs recovered after an additional 48 hours of infection. Unlike bacterial infections, where extracellular bacteria can be eliminated by treatment with gentamicin, there is no comparable compound that can selectively kill external parasites. We interpreted the absence of gp63 in EVs that are recovered following our protocol, to mean that external promastigote stage parasites, if still present, may contribute only a negligible number of particles to our EV preparation. We have included a bright field microscope image of infected cells at the time of culture medium recovery that shows that external parasites are absent.

Reviewer #2 (Comments to the Authors (Required)):

Comments to the author:

In this study, Gioseffi et al. investigate the composition of extracellular vesicles (EVs) released

from macrophages infected with Leishmania. The proteomic analyses demonstrated that infected macrophages release EVs with different compositions of proteins implicated in promoting vascular changes, such as Vash. They suggest that EVs from infected macrophages induce the release of angiogenesis molecules by endothelial cells and promotes epithelial cell migration and tube formation by endothelial cells in vitro. From these results, the authors propose that EV from infected macrophages promote vascularization in chronic Leishmania infections. Although the topic of this study is interesting and relevant to the disease, the conclusions are beyond what is shown by the data. The authors have used the appropriate methodology to isolate and physically characterize the EV (NTA and electron microscopy). However, the analyses to evaluate the role of EV in angiogenesis were only done in vitro, and whether they will have any role in chronic Leishmania infection is unknown.

Specific comments:

1) Structural concerns of the manuscript: The results section needs to be modified, and the combination of the results and discussion is not helpful in this manuscript. Even more important, the figures are of extremely low quality, some of them are unreadable, and the figure legends do not contain sufficient information. For example, the abbreviations used are not described, the statistical analysis and the number of replicates were not indicated.

We thank the reviewer for raising concerns about the structural presentation of our manuscript. In this submission, the manuscript layout has been revised. The results section and the discussion section are now separate. We apologize for the poor quality of the figures in the previous submission. The figures have been updated and should be of higher quality. The Figure legends have been edited by adding more information about abbreviations, statistical analysis, and number of replicates.

2) The purity of EV should be assessed.

This is an important issue. First, we made several new preparations of EVs, following the protocols described in the paper. It is important to highlight the fact that cultures are washed 3X with PBS to remove uninternalized parasites. Unfortunately, we cannot rule out the possibility that some parasites that are not internalized, remain stuck to macrophages. As we stated above, in response to Reviewer 1, We interpreted the absence of gp63 in EVs that are recovered following our protocol, to mean that external promastigote stage parasites, if still present, may contribute only a negligible number of particles to our EV preparation. We have included a bright field microscope image of infected cells at the time of culture medium recovery that shows that external parasites are absent. To confirm the purity of our LieEV and ceEV preparations, they were monitored for the presence of calnexin. Several studies have shown that calnexin is not a component of exosomes. Our analysis confirms this. As a control for protein loading, western blots were stained with Ponceau S. Samples from 3 separate isolations were analyzed and the results presented in the edited manuscript.

3) EVs have different membrane compositions, depending on the cell from which they have originated. The authors cannot rule out the possibility of EVs being released directly from *Leishmania*, instead of macrophage origin.

We agree with this statement from the reviewer. In response to Reviewer 1, we addressed the unlikely possibility that external parasites may contribute to EVs that are recovered following our protocol. If it indeed occurs, their contributions to the EV proteome are expected to be negligible.

Although it is known that *Leishmania* secrete exosomes, at this time we do not have markers that would differentiate parasite derived exosomes from host exosomes that are loaded with parasite molecules. We also cannot rule out the possibility that exosomes that are fully formed in internalized parasites, traffic through the macrophage, and are released to the extracellular milieu. Future studies will characterize the trafficking of parasite derived molecules within infected cells in greater detail.

4) Figure 3: The western blot for the GP63 band is not clear in the *Ldcen*^{-/-}. A loading control should be included.

In response to a suggestion by this reviewer, discussed below, we decided to remove this experiment from this resubmission.

5) Figure 4D: When closely examining the western blot, the *LdVash* 24h EV band is stronger than *LdVash* 72h and 96h. This result contrasts with the images of Figure 4A, which show an increase in the *LdVash* expression with the time. Consider quantifying the western blot bands.

The reviewer's observations are spot on. We are not certain why the mNG tagged proteins are less abundant in the Western blot analysis as compared to NTA. Your suggestion prompted us to quantify the bands in our Western blots.

6) Figure 5: The authors should use skin epithelial cells or spleen endothelial cells to test the effect of EV since these are the tissues related to *Leishmania* infection.

We understand the usefulness of this suggestion. We plan to include other cell lines from different tissues in future studies.

We should note that the Editor agrees that this issue is best handled at a different time (see Editor's comment above)

7) Figure 5: The cell lineage used in angiogenesis assays should preferably be murine since the assay involves EVs from the mouse.

We agree with the suggestion of the reviewer. However, the human cell lines are more widely used in these assays and so they were more readily available and easier to trouble shoot. Mouse cell lines will be used in future experiments.

We should note that the Editor agrees that this issue is best handled at a different time (see Editors comment above)

8) *Figure 5: CeEV Disrupted image in (A) is not a good representative figure of the graph (C).*

We thank the reviewer for pointing that out. The figure has been replaced.

9) *Figure 5: The status of angiogenesis molecules in the supernatant fluid from HUVEC cells incubated with disrupted EVs should be provided as a control.*

We understand the value of this suggestion. Unfortunately, at the time that these experiments were performed, the supernatants from the incubations with disrupted EVs were not saved.

10) *Figure 5E: It appears the positive control (VEGF) did not work.*

Indeed, these cells are maintained in a medium that contains VEGF; they apparently become unresponsive to the concentrations of VEGF that are suggested for these studies. It is also likely that the potency of our VEGF preparation had diminished. The response of the HUVEC cells to the other samples in these assays suggested that the poor response to VEGF may have been limited to that particular VEGF preparation.

11) *The authors do not formally investigate whether LiEV promotes vascularization in chronic Leishmania lesions since the analyses were done in vitro. Thus, their conclusions are not justified.*

The reviewer correctly points out that LiEVs were not evaluated on chronic infections. They were instead evaluated in surrogate assays that are widely used in studies of angiogenesis. We were careful to state in our conclusions that our findings suggest that LiEVs have the potential to promote these responses in *Leishmania* infections that are chronic. We have toned down our conclusions, as suggested by the Editor.

Minor comments:

1) *On what basis were 20 parasites per macrophage chosen?*

In initial experiments to develop the EV isolation protocol, infections for varying lengths of time were evaluated. To ensure that sufficient parasites were internalized when short term infections were evaluated, we settled on a 20:1 parasite to macrophage infection ratio. This

then became a part of our standard protocol even though we presently evaluate older infections.

2) Annexins expression levels are adjusted to the functional state of the cell. The discussion might benefit from some reflection on how Annexin A3 can be involved in macrophages infection and the EV release.

We agree with the reviewer that monitoring of Annexin A3 levels may provide greater insight into characteristics of the infection. With the change in the format of the manuscript we have included more of our rationale for monitoring Annexin A3 levels. Annexin A3 was rarely detected by mass spectrometry in the ceEV samples. The Western blot results in which up to 1×10^{10} particles per lane were analyzed, confirm that Annexin A3 is preferentially expressed in LieEVs samples as compared to ceEVs.

3) The Annexin A3 and GP63 data appears to have no connection with the main point of the paper.

As mentioned above, monitoring Annexin A3 can help confirm the mass spectrometry results. We agree with the concern that gp63 may not be connected to the main point of the paper. In this re-submission, in response to this reviewer's suggestion, we elected to not include the gp63 results in the figures.

4) "Protein concentrations determined by BCA agree with the NTA analyses that there is greater total protein content in the ieEV samples; however, ceEVs and ieEVs appear to contain equal protein content per particle." It is data not shown? The authors should discuss why ceEVs and ieEVs appear to contain equal protein content per particle.

After many experiments, we are still a bit unsure about the true link between the protein content as determined by BCA and the particle number per cell calculated from the NTA. We have elected to remove this statement as it distracts from the main points of the paper.

5) Overstate sentence: "Together, these studies demonstrated that there are impressive changes in vascularization of both visceral and cutaneous lesions that appear to be promoted by molecules that are released from infected macrophages in infected tissues." The mentioned papers (Horst et al, 2009, Weinkopff et al, 2016, Yurdakul et al, 2011 and Dalton et al, 2015) demonstrated that the molecules were released by macrophages present in infected mice. None of these papers has shown that infected macrophages release the molecules.

We thank the reviewer for this insightful and critical analysis of the studies in the literature that have commenced dissecting the mechanisms that underly changes in vascularization of *Leishmania* infections. We have changed that statement in the manuscript. We have included

a more critical statement that is consistent with your suggestion.

6) *Figure 4: The results section is different from the figure order. The description of Figure 4B in the text is referring to 4C in the figure. The authors state that after infection with the recombinant parasites there was an increase in the total number of EVs over time but there is no statistical test information in the figure and the legend.*

We thank the reviewer for picking up this error in the manuscript. The figure legend and the text that describes this figure has been changed accordingly.

Reviewer #3 (Comments to the Authors (Required)):

Leishmania donovani is the causative agent of a potentially fatal visceral infection. In this manuscript, Gioseffi, et al. explored the composition and role of extracellular vesicles (EVs) released from infected macrophages. Leishmania infected macrophage EVs (LiEVs) contain several host proteins that can cause endothelial cells to migrate, engage in tube formation and release angiogenesis promoting factors such as IL-8, GCSF/CSF-3 and VEGF-A. Additionally, more than 50 L. donovani - derived proteins were identified in LiEVs, including a homolog of mammalian Vasohibins (LdVash), an angiogenesis promoting mediator. Taken all together, these results show that L. donovani infection can alter the composition of LiEVs and plays a role in vascularization during chronic disease. This is one of the first reports on leishmanial proteins found in host exosomes, and the first on L. donovani. The manuscript has good logical experimental flow and discussion of the results and implications of the paper. Minor issues are highlighted below:

We thank the reviewer for this truly clear understanding of the novelty of our findings.

Minor:

1. The present studies were performed using RAW264.7 macrophages. The authors should include a brief explanation about why they chose immortalized cells over primary cells such as bone marrow derived macrophages, which are more physiologically relevant

We thank the reviewer for raising this issue. In light of the novelty of these experiments, it was necessary that we use a cell line that offered uniformity and that we could scale up without sacrificing a lot of animals. The composition of EVs, is cell type specific. The use of a cell line helps to ensure the rigor of our experiments, With our new understanding of EV composition and the dynamics of EV release from infected cells, we are presently better equipped to perform similar experiments on primary cells, which we agree, are more relevant in the studies of a chronic infection. We have included the rationale for using RAW264.7 cells in the Materials and Methods section (Mammalian cell culture).

2. The authors mention Hassani and Oliver 2013 (PMID: 23658846). It would be interesting to discuss and draw a parallel between the proteins found in *L. donovani* and those previously found by Hassani and Oliver in *L. mexicana*-derived exosomes

In that study, Hassani and Olivier report finding only a single parasite derived protein – gp63 in exosomes preparations from infected cultures. That study was the first to show that a *Leishmania*-derived protein is included in the exosomal cargo from infected cells. In the manuscript, we acknowledge their finding and discuss the biology of gp63, which makes it a unique *Leishmania* molecule.

3. Figures and Tables:

a. Table 1 cannot be found in the manuscript

We apologize that this table wasn't readily available in the initial submission. Table 1 is in the supplement.

b. Include a loading control in the western blots of figure 2B, 3B and 4D

Given that there are no established molecules in EVs that are invariant, we elected to include a representative Ponceau S stain of the blots to demonstrate that comparable amounts of protein were loaded. In light of differences in protein content of exosomes and lysates, this is the best approach that we know of to demonstrate protein loading levels.

c. Part of the content of Figure 2 and 3 might be better represented into tables, consider separating them up

We agree with the reviewer. We have separated Figure 2 and 3 into Tables 1 and 2.

d. In the legend of Figure 5 there should be a closed parenthesis after "E" instead of a period

Thank you. We have included the change.

4. Minor grammatical errors:

a. In the result section *Differential host protein composition in LiEVs* it says: "low abundant proteins". The word "abundant" should be substituted with "abundance"
Change made

b. In the result section *Extracellular vesicles derived from Leishmania-infected cells activate angiogenesis* there should be a period between "cell tube formation" and "Cell migration"

Change made

c. All the "et al." have to be italicized throughout the paper

Done.

5. Please consider moving part of *Isolation and characterization of extracellular vesicles released from Leishmania donovani infected RAW264.7 macrophages to the methods section*

Done.

October 1, 2020

RE: Life Science Alliance Manuscript #LSA-2020-00742-TR

Dr. Peter E Kima
University of Florida
Department of Microbiology and Cell Science University of Florida Building 981, Box 110700
Gainesville, FL 32611

Dear Dr. Kima,

Thank you for submitting your revised manuscript entitled "Leishmania-infected macrophages release extracellular vesicles that can promote lesion development". We would be happy to publish your paper in Life Science Alliance pending final revisions necessary to meet our formatting guidelines.

Along with the formatting requests below, please also attend to the following points:

- please add an ORCID ID for the corresponding author-you should have received instructions on how to do so
- please use the [10 author names, et al.] format in your references (i.e. limit the author names to the first 10)
- please add your supplementary figure legends to the main manuscript text right after your figure legends for your main figures
- please add scale bars to Fig. 4A; Fig. 5A,D; Fig. S1; Fig S4A
- please add a callout for Fig. S3A in your main manuscript text
- please rename the section 'Methods' to 'Materials & Methods'
- Please deposit the Mass Spec data in an online repository and provide the accession number in the revised manuscript (See Data Deposition guidelines here - <https://www.life-science-alliance.org/manuscript-prep#data-depot>)
- Please provide original unedited source data for the blots shown in Figures 2B,C, Figure 4D and Figure S5

A. FINAL FILES:

B. MANUSCRIPT ORGANIZATION AND FORMATTING:

Sincerely,

Shachi Bhatt, Ph.D.
Executive Editor
Life Science Alliance

Reviewer #1 (Comments to the Authors (Required)):

I believe that the additional data and the enhanced quality of the figures further strengthens this strong manuscript. I appreciate their explanation for the difficulty in identifying markers to distinguish host-derived vs Leishmania-derived vesicles. In any case, the phenotype of these vesicles is clear. I have no additional concerns.

Reviewer #3 (Comments to the Authors (Required)):

All my concerns are addressed

October 8, 2020

RE: Life Science Alliance Manuscript #LSA-2020-00742-TRR

Dr. Peter E Kima
University of Florida
Department of Microbiology and Cell Science University of Florida Building 981, Box 110700
Gainesville, FL 32611

Dear Dr. Kima,

Thank you for submitting your Resource entitled "Leishmania-infected macrophages release extracellular vesicles that can promote lesion development". It is a pleasure to let you know that your manuscript is now accepted for publication in Life Science Alliance. Congratulations on this interesting work.

DISTRIBUTION OF MATERIALS:

Again, congratulations on a very nice paper. I hope you found the review process to be constructive and are pleased with how the manuscript was handled editorially. We look forward to future exciting submissions from your lab.

Sincerely,

Shachi Bhatt, Ph.D.
Executive Editor
Life Science Alliance
<https://www.life-science-alliance.org/>
Tweet @SciBhatt @LSAJournal